

# AERO-MAP: A data compilation and modelling approach to understand spatial variability in fine and coarse mode aerosol composition

Natalie M. Mahowald[1], Longlei Li[1], Julius Vira[2], Marje Prank[2], Douglas S. Hamilton[3], Hitoshi Matsui[4], Ron L. Miller[5], Louis Lu[1], Ezgi Akyuz[6], Daphne Meidan[1], Peter Hess[7], Heikki Lihavainen[8], Christine Wiedinmyer[9], Jenny Hand[10], Maria Grazia Alaimo[11], Célia Alves[12], Andres Alastuey[13], Paulo Artaxo[14], Africa Barreto[15], Francisco Barraza[16], Silvia Becagli[17], Giulia Calzolai[17], Shankararaman Chellam[18], Ying Chen[19], Patrick Chuang[20], David D. Cohen[21], Cristina Colombi[22], Evangelia Diapouli[23], Gaetano Dongarra[11], Konstantinos Eleftheriadis[23], Johann Engelbrecht[24], Corinne Galy-Lacaux[25], Cassandra Gaston[26], Dario Gomez[27], Yenny González Ramos[28,15], Roy M. Harrison[29], Chris Heyes[30], Barak Herut[31,32], Philip Hopke[33,34], Christoph Hüglin[35], Maria Kanakidou[36,37,38], Zsofia Kertesz[39], Zbigniew Klimont[30], Katriina Kyllönen[2], Fabrice Lambert[40,41], Xiaohong Liu[42], Remi Losno[43], Franco Lucarelli[17], Willy Maenhaut[44], Beatrice Marticorena[45], Randall V. Martin[46], Nikolaos Mihalopoulos[35,47], Yasser Morera-Gomez[48], Adina Paytan[49], Joseph Prospero[25], Sergio Rodríguez[50,15], Patricia Smichowski[27], Daniela Varrica[11], Brenna Walsh[46], Crystal Weagle[46], Xi Zhao[42]

[1]Department of Earth and Atmospheric Sciences, Cornell University, Ithaca, NY, 14853, USA

[2] Finnish Meteorological Institute, Helsinki, Finland

[3] Department of Marine, Earth and Atmospheric Sciences, North Carolina State, Raleigh, NC, USA

[4] Graduate School of Environmental Studies, Nagoya University, Nagoya, Japan 464-8601

[5] National Aeronautics and Space Administration, Goddard Institute for Space Studies, Columbia University, NY, NY 10025

[6] Eurasia Institute of Earth Sciences, Istanbul Technical University, 34467 Istanbul, Turkey

[7] Department of Biological and Environmental Engineering, Cornell University, Ithaca NY, USA

[8] SIOS Knowledge Centre, Postboks 156, 9171 Longyearbyen, Norway

[9] Cooperative Institute for Research in Environmental Sciences at the University of Colorado Boulder, Boulder, CO, USA

[10] Cooperative Institute for Research in the Atmosphere, Colorado State University, Fort Collins, CO, USA,

[11] Dip. Scienze della Terra e del Mare, University of Palermo, Italy

[12] Centre for Environmental and Marine Studies (CESAM), Department of Environment, University of Aveiro, 3810-193, Aveiro, Portugal

[13] Institute of Environmental Assessment and Water Research (IDAEA-CSIC), 08034, Barcelona, Spain

[14] Instituto de Fisica,Universidade de Sao Paulo, 05508-090, Sao Paulo, SP, Brazil

[15] Izaña Atmospheric Research Center (IARC), Agencia Estatal de Meteorología (AEMET), Santa Cruz de Tenerife, Spain



[16] Saw Science, Invercargill, New Zealand
[17] Department of Physics and Astronomy, Universita di Firenze and INFN-Firenze, 50019 Sesto Fiorentino, Italy
[18] Department of Civil & Environmental Engineering, Texas A&M University, College Station, TX 77843-3136, USA
[19] Dept. Environ. Sci. Engr. Fudan University Jiangwan Campus 2005 Songhu Road, Shanghai, China
[20] Earth & Planetary Sciences Department, Institute of Marine Sciences, University of California, Santa Cruz, CA, 95064 ,
USA.
[21] Centre for Accelerator Science, Australian Nuclear Science and Technology Organisation, New Illawarra Rd, Lucas
Heights, NSW, Australia
[22] Environmental Monitoring Sector, Arpa Lombardia, Via Rosellini 17, 20124 Milan, Italy
[23] Environmental Radioactivity & Aerosol Technology for Atmospheric & Climate impact Lab, INRaSTES, N.C.S.R.
Demokritos, 15341 Ag. Paraskevi, Attiki, Greece
[24] Desert Research Institute (DRI), 2215 Raggio Parkway, Reno, Nevada 89512-1095
[25] Laboratoire d Aerologie, Universite de Toulouse, CNRS, Observatoire Midi Pyrenees, Toulouse, France
[26] Rosenstiel School of Marine and Atmospheric Science, University of Miami, Miami, FL, 33149, US
[27] Comision Nacional de Energia Atomica, Gerencia Química, Av. Gral Paz 1499, B1650KNA, San Martin, Buenos Aires,
Argentina
[28] Scientific Department, CIMEL, Paris, France.
[29] School of Geography, Earth and Environmental Sciences, University of Birmingham, Edgbaston, Birmingham B15 2TT,
United Kingdom
[30] Energy, Climate and Environment Program, International Institute for Applied Systems Analysis, 2361 Laxenburg,
Austria
[31] Israel Oceanographic & Limnological Research, Tel Shikmona, Haifa, 31080, Israel
[32] University of Haifa, Haifa, 3103301, Israel
[33] Clarkson University, Potsdam, NY, USA,
[34] Department of Public Health Sciences, University of Rochester School of Medicine and Dentistry, Rochester, NY, USA,
[35] Swiss Federal Laboratories for Materials Science and Technology(EMPA), CH-8600 Duebendorf, Switzerland
[36] Environmental Chemical Processes Laboratory (ECPL), Department of Chemistry, University of Crete, Heraklion, Greece.
[37] Center of Studies of Air quality and Climate Change, Institute for Chemical Engineering Sciences, Foundation for
Research and Technology Hellas, Patras, Greece.
[38] Excellence Chair, Institute of Environmental Physics, University of Bremen, Bremen, Germany
[39] HUN-REN Institute for Nuclear Research (ATOMKI), Debrecen, Hungary
[40] Geography Institute, Pontificia Universidad Catolica de Chile, Santiago, 7820436, Chile
[41] Center for Climate and Resilience Research, Santiago, Chile
[42] Department of Atmospheric Sciences, Texas A&M University, College Station, TX 77843
[43] Institut de Physique du Globe de Paris, Universite de Paris, Paris, France
[44] Department of Chemistry, Ghent University, Gent, Belgium
[45] Laboratoire Interuniversitaire des Systemes Atmospheriques (LISA), Universit«es Paris Est-Paris Diderot-Paris 7, UMR
CNRS 7583, Cr«eteil, France
[46] Energy, Environmental and Chemical Engineering, Washington University, St. Louis, MO, USA.
[47] Institute for Environmental Research and Sustainable Development, National Observatory of Athens, Pendeli, Greece
[48] Universidad de Navarra, Instituto de Biodiversidad y Medioambiente BIOMA, Irunlarrea 1, 31008, Pamplona, España



[49] Earth and Planetary Science, University of California, Santa Cruz, CA, USA
[50] Consejo Superior de Investigaciones Científicas, IPNA CSIC, Tenerife, Canary Islands, Spain.

*Correspondence to*: Natalie M. Mahowald (mahowald@cornell.edu)
**Abstract.** Aerosol particles are an important part of the Earth system, but their concentrations are spatially and temporally
heterogeneous, as well as variable in size and composition. Particles can interact with incoming solar radiation and outgoing
long wave radiation, change cloud properties, affect photochemistry, impact surface air quality, change the surface albedo of
snow and ice, and modulate carbon dioxide uptake by the land and ocean. High particulate matter concentrations at the
surface represent an important public health hazard. There are substantial datasets describing aerosol particles in the
literature or in public health databases, but they have not been compiled for easy use by the climate and air quality modelling
community. Here we present a new compilation of $PM_{2.5}$ and $PM_{10}$ aerosol observations, focusing on the spatial variability
across different observational stations, including composition, and demonstrate a method for comparing the datasets to
model output. Overall, most of the planet or even the land fraction does not have sufficient observations of surface
concentrations, and especially particle composition to understand the current distribution of particles. Most climate models
exclude 10-30% of the aerosol particles in both $PM_{2.5}$ and $PM_{10}$ size fractions across large swaths of the globe in their current
configurations, with ammonium nitrate and agricultural dust aerosol being the most important omitted aerosol types.

## 1 Introduction

Intergovernmental Panel on Climate Change (IPCC) reports and studies have highlighted the role of uncertainties in human-
induced changes to aerosol concentration and composition in the atmosphere in limiting our ability to project future climate
(IPCC, 2021; Gulev et al., 2021; Szopa et al., 2021). Aerosol particles are also a major contributor to air quality problems,
which reduce life expectancy and quality of life (Burnett et al., 2018). Aerosol particles are suspended liquids or solids in the
atmosphere originating from diverse sources and composed of a wide variety of chemicals (e.g., sea salts, dust, sulfate,
nitrate, black carbon, organic carbon). Particles interact with incoming solar radiation, outgoing long wave radiation, change
cloud properties and lifetimes, and modify atmospheric photochemistry (Mahowald et al., 2011; Kanakidou et al., 2018;
Bellouin et al., 2020). Once deposited on the surface, they can modify land and ocean biogeochemistry, as well as the
albedo of snow and ice surfaces (Mahowald et al., 2017; Hansen and Nazarenko, 2004; Skiles et al., 2018). New satellite
remote sensing measurements provide important information about temporal and spatial distribution of aerosol particles, but
challenges remain in quantifying the size and chemical composition of aerosol (Kahn et al., 2005; Tanré et al., 1997; Remer
et al., 2005). In addition, the AERONET surface remote sensing network provides some information about loading, size and





absorbing aerosol properties related to composition (Holben et al., 1991; Dubovik et al., 2002; Schuster et al., 2016;
Gonçalves Ageitos et al., 2023; Obiso et al., 2023). Both the magnitude of the effects, and sometimes the sign of the aerosol
effects on climate are dependent on the composition and size of particles (Mahowald et al., 2011, 2014a; Bond et al., 2013;
IPCC, 2021). In addition, one cannot understand the impact of humans on aerosol particles without understanding the
sources of particles, which determines their chemical composition. Obtaining information about the composition and size of
particles in many cases requires in situ observations, which are limited in space and time (Hand et al., 2017; Philip et al.,
2017; Yang et al., 2018; Collaud Coen et al., 2020).
The climate and aerosol modelling community, especially under the auspices of AEROCOM, has compiled datasets and
organized comparison projects that have provided substantial information to improve aerosol models (Huneeus et al., 2011;
Textor and others, 2006; Dentener et al., 2006; Schulz et al., 2006; Gliß et al., 2021) or knowledge of the aerosol properties
like cloud condensation nuclei (Laj et al., 2020; Fanourgakis et al., 2019). However, most of these comparisons include data
only from North America and Europe (e.g., Szopa et al., 2021). In addition, previous compilation studies have focused
primarily on understanding fine aerosol particles (here defined as particle with a diameter less than 2.5 µm) and improving
model simulation of these particles, because of their importance for air quality, cloud interactions and short-wave forcing
(Collaud Coen et al., 2020; Bellouin et al., 2020; Fanourgakis et al., 2019). Coarse mode particles (defined as those particles
with a diameter larger than 2.5 µm) are important for long wave radiation interactions, cloud seeding and for
biogeochemistry, but these interactions have received less attention (Jensen and Lee, 2008; Mahowald et al., 2011; Karydis
et al., 2017; Chatziparaschos et al., 2023). In contrast to the many fine aerosol compilations and comparisons (usually
considering particles with diameter less than 2.5 µm or $PM_{2.5}$), there are fewer studies focusing on aerosol compilations for
both fine and coarse particles, and their comparison to models (Kok et al., 2014b; Albani et al., 2014b; Huneeus et al., 2011;
Gliß et al., 2021; Kok et al., 2021). Nonetheless, there are many observations of the coarse particle mass included in the
particles with diameter less than 10 µm ($PM_{10}$) (e.g., Hand et al., 2017), and most climate models include these particles
(e.g., Huneus et al., 2011). Compilations of in situ data are available for dust and iron particles (Kok et al., 2014b; Albani et
al., 2014b; Mahowald et al., 2009) and for sea salts (Gong et al., 1997). Other studies have focused on the important topics
of wet deposition (Vet et al., 2014) or trends in aerosol properties (e.g., AOD, surface PM) (Mortier et al., 2020; Aas et al.,
2019). Observations of $PM_{10}$ or coarse and fine particles are available for many regions and individual sites (e.g., Malm et
al., 2007; Hand et al., 2019; Maenhaut and Cafmeyer, 1998; Artaxo and Maenhaut, 1990; McNeill et al., 2020) but have not
previously been compiled into one database. Aerosol modelers need as much information as possible about the composition
of the particles. Thus, there is a need to compile both $PM_{2.5}$ and $PM_{10}$ in situ concentration data into one database to make it
easy for modellers to compare model results with observations. One goal the aerosol community should work towards is
making aerosol measurement datasets publicly available, while acknowledging the principal investigators who produced
these datasets, which we hope this paper serves as a step towards achieving.



The current generation of Earth system models used for the IPCC simulations tends to include the dominant aerosol
particles (desert dust, sea spray, black carbon (BC), organic matter (OM) and sulfate) but not all particles. For example,
some Earth system models ignore ammonium nitrate particles although these are known to be important for climate and
biogeochemistry, and are impacted by human activities (Paulot et al., 2016; Adams et al., 1999; Thornhill et al., 2020). In
addition, some models focus only on fine mode OM and BC particles, although there is evidence for coarse mode particles of
both carbonaceous particles (Graham et al., 2003; Mahowald et al., 2005).  Agricultural or land use sources of dust are not
included in most models, although they could represent 25% of the anthropogenic sources (Ginoux et al., 2012), and
significantly impact transported transhemispheric aerosol composition (García et al., 2017). In addition, fugitive, combustion
and industrial dust emissions have traditionally been ignored as well, although emission datasets are available (Philip et al.,
2017).  In this study we use available observations to constrain a model estimate of the total $PM_{10}$ and $PM_{2.5}$, and deduce the
importance of these often-neglected aerosol particles.  We propose a method for comparing particles that are not directly
measured (dust or sea salts) using their elemental composition. Note that we exclude super coarse ($>PM_{10}$) particles here
because of the lack of available data, although studies have suggested their importance for climate interactions (e.g., Adebyi
et al., 2023).
Aerosols are highly hetereogeneous in space and time: here we focus on characterizing in observations and models the
spatial variability of the surface concentrations, as it is arguably the largest, spanning 4-5 orders of magnitude (e.g.,
Mahowald et al., 2011; Section 3.2). Spatial variability in surface concentrations is one of the least well known, for example
in the many unmonitored regions of the globe (e.g., Szopa et al., 2021).  Understanding spatial variability in aerosols, and the
composition of aerosols is key to understanding how aerosols have evolved in the past, and how they will evolve in the
future, as some regions are dominated by fossil fuel derived aerosols, which may have peaked in magnitude, while other
regions aerosols are driven by agriculture or by natural aerosols (Turnock et al., 2020; Kok et al., 2023).  In addition,
different aerosols have different impacts on climate, for example, knowing whether aerosols are scattering or absorbing
changes the sign of the interaction (Li et al, 2022).  Some aerosols also serve as better cloud or ice nuclei than others, while
biogeochemical impacts are very sensitive to composition (Mahowald et al., 2011).  Knowing the order of magnitude even in
regions with aerosols (e.g., contrasting 0.1 to 0.001) is important for aerosol cloud interactions that can be non-linear
especially at low aerosol levels (Carslaw et al., 2013). While remote sensing data can provide important information about
high aerosol load regions, there is only limited information about the composition (e.g., single scattering albedo under very
high aerosol optical depth (AOD>0.4) conditions, for example (Dubovik et al., 2000)).  We focus on the spatial distribution
of climatological mean, as that is easily obtained from models, and the most important variable for many climate impacts
like radiative effects or aerosol-cloud interactions except in cases with large infrequent events (e.g., Clark et al., 2015;
Fasulo et al., 2022).  The climatological mean is obviously less important for extreme air quality events, or for understanding
temporal trends or pollution events, and thus other datasets should be developed for these attributes (e.g., Bowdalo et al.,



2024). There have been trends in emissions especially of anthropogenic aerosols over the last 40 years (Quass et al., 2022;
Bauer et al., 2022), which we do not access in this study.

For this study we focus on the following: a) compiling climatologically averaged means of available $PM_{2.5}$ and

$PM_{10}$ aerosol data, including aerosol composition into a new publicly available database for the modelling community
(AEROMAP) across as much of the globe as possible; b) presenting a methodology to compare these observations to an
Earth system model; c) identifying the measurement and modelling gaps from this comparison. In this paper, we focus on
the climatological average spatial distribution of aerosol particles and key chemical composition information.
**2 Description of Methods**
**2.1 Observational data**
PM observations are made by multiple networks, or during specific field campaigns, and for different size cut-offs, with and
without a description of chemical composition. Data was collected by advertising at international meetings (Wiedinmyer et
al., 2018), searching the literature, contacting principal investigators and accessing publicly available datasets. As expected,
most of the observations are over North America or Europe, with much of the rest of the land areas and most of the ocean
much more poorly observed (Fig. 1; Supplemental dataset 1). For this study, we include both $PM_{2.5}$ and $PM_{10}$ daily (or
multiple day averages) data sets that were made available by the investigators or are available from public web sites (Fig. 1;
supplemental dataset 1). Some measurement sites measure $PM_{2.5}$ and coarse ($PM_{2.5}$ to $PM_{10}$) aerosols. For those sites, we
convert the latter to $PM_{10}$ for comparison. Some measurement sites have only a few observations of composition or mass,
while others have multiple years: we included less complete datasets at sites in regions with limited data. In some poorly
measured regions, we include total suspended particles (TSP) datasets. The time period for different datasets is included in
the supplemental dataset 1.
Detailed studies have shown that $PM_{10}$ and $PM_{2.5}$ samplers can differ in the sharpness of their size cut-off (Hand et al.,
2019). As an example, comparisons between data from the U.S. Environmental Protection Agency (EPA) Federal Reference
Method sites and data from the Interagency Monitoring of Protected Visual Environments (IMPROVE) network show that
the coarse matter from collocated sites in both networks were offset by 28% (Hand et al., 2019). There was a bias when data
were compared (slope of 0.9), but the correlation coefficient was high (0.9) suggesting overall a good agreement. We focus
here on surface station measurements of $PM_{10}$ and $PM_{2.5}$, since our model and most models only consider mass up to $PM_{10}$.
For that reason, our model deposition is not directly comparable to observational bulk/total atmospheric deposition since
larger particles may dominate the deposition close to the source areas (Kok et al., 2017; Mahowald et al., 2014b; Neff et al.,
2013). Measuring absolute dry and wet deposition rates is also technically more challenging (especially dry deposition, since





the particles can be re-entrained into the atmosphere), but worthwhile (Heimburger et al., 2012; Prospero et al., 1996). In
regions with little data (e.g., outside of North America and Europe) we include measurements of total suspended particulates
(TSP) with the PM$_{10}$, because of the lack of size-resolved data.  Data from the Japanese air quality network use a different
inlet for the PM10 cutoff as well, which will include a slightly larger size fraction (https://tenbou.nies.go.jp/download/).
In addition to particulate matter in the PM$_{10}$ and PM$_{2.5}$ size fractions, we also compile the following observations to compare
to the model: black carbon (BC), elemental carbon (EC), organic carbon (OC) (or particulate organic material, OM, that is
here considered to be 1.8 x OC in mass), sulfate, nitrate, aluminum, sodium and chloride.  To include both BC (based on
light absorption measurements) and EC (based on thermal oxidation induced combustion measurements) data are also a
source of uncertainty, both are proxies of the soot combustion particles since they are based on different measurements
techniques, and there is no accepted equivalence between them (Mbengue et al., 2021). Details on how the model is
compared to data for different elements is in section 3.2.
For this paper, we focus on the climatological annual means for 1986-2023 which are calculated for all values at each station
that are above the detection limit and reported here. At some stations or times, concentrations can be below the detection
limit, and excluding these data or time periods could bias our average values.  We focus on the stations that have more than
50% of the data above the detection limit, and exclude other sites. For those included stations, if the values were reported as
below the detection limit, we include in the average one-third of the minimum detection limit. The reported detection limits
should bound the upper limit of aerosol mass and allow us to include sites, whose observations were otherwise too low to
include, while reducing the potential biasing of our compilation towards higher values (Supplemental dataset 1).
**2.2 Model description**
Simulations of aerosol particles were conducted using the aerosol parameterizations within the Community Atmosphere
Model, version 6 (CAM6), the atmospheric component of the Community Earth System Model (CESM) developed at the
National Center for Atmospheric Research (NCAR) (Hurrell et al., 2013; Scanza et al., 2015; Liu et al., 2012). The aerosol
module in this version is closely related to the module used in the Energy Exascale Earth System Model (Golaz et al., 2019;
Caldwell et al., 2019).  Simulations were conducted at approximately 1°x1° horizontal resolution with 56 vertical layers for
four years, with the last three years (2013-2015) used for the analysis (Computational and Information Systems Laboratory,
2019). The model simulates three-dimensional transport and wet and dry deposition for gases and particles based on
MERRA2 winds (Gelaro et al., 2017).
The model included prognostic dust, sea salts, BC, OM, and sulfate particles in the default version, using a modal scheme
based on monthly mean emissions for the year 2010 (Liu et al., 2012, 2016; Li et al., 2021). For this study, the coarse size
mode (mode 3) was returned to the CAM5 size parameters (geometric standard deviation of 1.8) to better simulate coarse



mode particles, and improve the dry deposition scheme and optics used in the model for simulating coarse mode particles
like dust as described in Li et al. (2022b).
Desert dust is entrained into the atmosphere in dry, sparsely vegetated regions subject to strong winds. We use the Dust
Entrainment and Deposition scheme (Zender et al., 2003) with the emitted size distribution given by the updated Brittle
Fragmentation Theory (Kok et al., 2014b, a) with improved incorporation of aspherical particles for optics and deposition (Li
et al., 2022b; Huang et al., 2021; Kok et al., 2017). Fossil fuel and natural emissions of sulfate, OM, and BC follow the
Climate Model Intercomparison Project 6 historical data for 2010 (Gidden et al., 2019).

### 2.2.1 Modelling of additional aerosol sources and types

The model was modified to allow the addition of several new aerosol particles based on codes with expanded dust speciation
(Li et al., 2022b) but here the extra dust tracers are used for the additional species as described below. The additional
sources of particles use the same optical properties as bulk dust for this sensitivity study. The following particles were
added, and the amount of emissions in each the $PM_{2.5}$ and $PM_{10}$ sizes and contribution to surface concentration and aerosol
optical depth are shown in Table 1. In addition, some of the base case aerosol emissions were modified to match
observations, as discussed below.
Agricultural sources of dust are added to this version of the model using the same emission scheme as for natural sources
(Kok et al., 2014b, a; Li et al., 2022b), but applied to the crop area, and each region is tuned to have the percentage amount
of anthropogenic dust to match satellite based observations (Ginoux et al., 2012), except Australia, where other estimates
(Bullard et al., 2008; Mahowald et al., 2009; Webb and Pierre, 2018) suggest a lower amount (see Table S1 for comparisons,
based on Brodsky et al., 2023). Agricultural dust is separately considered by the model, so its importance can be evaluated.
Coarse BC and OC as well as fine and coarse ash from industrial sources were added. Emissions estimated from the GAIN
model are added to the model using the ECLIPSEV6_CLE base case (Klimont et al., 2017; Philip et al., 2017). Coarse BC
and OM from biomass burning were assumed to be 20% of the fine mode mass (Mahowald et al., 2005).
Primary biogenic particles are released from ecosystems either as integral particles, such as bacteria, pollen or spores, or as
accidentally entrained leaf pieces (Jaenicke, 2005; Mahowald et al., 2008; Despres et al., 2012; Burrows et al., 2009; Heald
and Spracklen, 2009). These sources are poorly observed or understood, and thus looking at coarse mode organic material in
this study could provide additional constraints on the budget. Assumptions about size are likely to be very important for the
resulting distribution and impacts, e.g., studies show that P budgets are quite different if 5 different size bins or 1 size bin are
included in models (Brahney et al., 2015). Four different types of primary biogenic particles were included: bacteria, spores
and other miscellaneous emissions (leaf bits, pollen, etc.) from land ecosystems, as well as a marine organic aerosol.



Included bacteria sources were read in from a monthly climatology (Burrows et al., 2009). Spore sources were calculated
offline and read into the model based on observed leaf area index, temperature, and a source parameterization (Janssen et al.,
2020; Heald and Spracklen, 2009). Other terrestrial emissions were estimated based on leaf area index following Mahowald
et al. ( 2008). Marine organic aerosol emissions were included based on the physically based scheme OCEANFILMS
(Burrows et al., 2014). Marine organics are externally mixed with sea spray, following Zhao et al. (2021). OCEANFILMS
only estimates the fine mode organic mass, and here we assume that the coarse mode marine organic mass equals 1% of the
seaspray mass (Gantt et al., 2011). The assumptions about the mass and fraction in each size bin are shown in Table 1.
Ammonium nitrate aerosol particles are not included in the standard CAM6 nor in E3SM, but are thought to be important for
aerosol optical depth and surface concentrations (Paulot et al., 2016; Adams et al., 1999; Thornhill et al., 2020; Bauer et al.,
2007, 2016). Nitrate can also react with dust particles, for example, but that is ignored in this study (Wolf, 2006; Dentener et
al., 1996). Ammonium nitrate particles require tropospheric chemistry interactions because the nitrogenous based particles
are both a source and a sink for gaseous nitrogen species, which are key elements of tropospheric photochemistry and the
particles are in chemical equilibrium with the gas phase (e.g., Nenes et al., 2021; Baker et al., 2021; Bauer et al., 2007;
2016), so simulations using the CAM-CHEM model with tropospheric photochemistry are used covering the same time
period (Vira et al., 2022). Simulations with chemistry were conducted at 2°x2° resolution and are linearly interpolated to
1°x1° resolution used for the other modelled particles. Sulfate in the CAM6 is assumed to be in the form of ammonium
sulfate and the nitrate is assumed to be in the form of ammonium nitrate for these studies, so as a rough approximation only
the ammonium nitrate needs to be added to consider nitrogenous aerosol optical depth. While aerosol amounts are
simulated, ammonium nitrate aerosol optical depth is not calculated within the model but offline. The model does calculate
sulfate aerosol optical depth, which has a roughly similar increase in size with humidity, and similar optical properties as
long as the nitrates and sulfates are in similar size fractions (Paulot et al., 2016; Bellouin et al., 2020). Therefore the aerosol
optical depth from ammonium nitrate (per unit mass) is assumed to be proportional to the sulfate aerosol optical depth per
unit mass in each grid box at each time interval. Detailed comparison of the nitrate and ammonia particles, and other species
was conducted in Vira et al. (2022). Overall, the model can simulate some of the spatial distribution, but overestimates the
nitrate aerosol amounts. This is also seen in Vira et al. (2022), and as shown in Table 1, the calculated nitrate aerosol
amounts are multiplied by 0.5 to best match the available observations.
**2.3 Model-observation comparison methodology**
Comparisons of the observations to model concentrations were done using BC, OC, $SO_4^{2-}$, Al, $NO_3^-$, $NH_4^+$, Na, and Cl
composition measurements. Some of these elements/compounds map directly onto model constituents (BC, OC, $SO_4^{2-}$, $NO_3^-$
, and $NH_4^+$), while others serve as proxies for modelled constituents (Al for dust and industrial ash, Na and Cl for sea salts, S
for sulfate, etc.). We use non-sea-salt sulfate in ocean regions for estimating sulfate. Some observing networks like



IMPROVE use a composite of elements to deduce dust amounts (e.g., Hand et al., 2017). We do not choose to do this for
two reasons: 1) at some sites not all the elements are available, and 2) because these elements derive not only from desert
dust, but also from industrial sources.  Instead, here we explicitly include industrial ash sources and the resulting Al. Note
that model values come from the midpoint of the bottom level of the model (~30 m) while the observations are usually taken
at 2 or 10 m high. There are several sources of measurement differences between different networks as well as between
model and observations.  Modelled values of PM content, which assume dry particles, are used, while gravimetric
measurements in some networks are equilibrated at 50% relative humidity, thus 5-25% of the mass of measured PM can be
water (Prank et al., 2016; Burgos et al., 2020). In addition, comparisons of coarse composition mode composition at co-
located sites in the US show that the inlet type can cause ~30% difference in measured mass (Hand et al., 2017).
For the most part, we use model output for which there is a on- to-one relationship with what is being measured (BC, sulfate,
etc).  However, for dust this is not straightforward, as dust is composed of multiple elements.  Here we use Al as a proxy for
dust, as it is relatively constant (~7%) in dust (as opposed to Ca, which varies highly, or Fe which varies moderately) (Zhang
et al., 2015).  Al sources are primarily from dust, agricultural dust, road dust and industrial ash emissions; we ignore minor
emissions from volcanoes, marine sea spray and primary biogenics for this study (Mahowald et al., 2018).  Assumptions
about the model composition and how they are compared to observations are shown in Table S2.  For example, OM is
assumed to be 1.8 times OC.
Harmonizing models with different types of measurements is critical (Huang et al., 2021). Models operate with the
geometric or aerodynamic particle diameter, whereas in practise the measurements are done with a variety of particle
equivalent diameter, e.g., optical, volume equivalent, projected-area equivalent, aerodynamic diameter or electrical mobility
diameter, depending on the instrument used (Hinds, 1999; Reid et al., 2003; Rodríguez et al., 2012). In the inlets of the
samplers used for the mass-measurements and collection of $PM_{2.5}$ and $PM_{10}$ particles for subsequent chemical analysis, such
size cut-off at 2.5 µm and 10 µm is defined in terms of aerodynamic diameter (i.e., Stokes diameter (involving size and
shape) weighed by the square root of the particle density; Hinds, 1999)_ The sharpness of the cut-off of such inlets
influences the $PM_{2.5}$ and $PM_{10}$ mass concentration (Hand et al., 2019; Wilson et al., 2002). The $PM_{10}$ size cut-off
aerodynamic diameter is equivalent to $PM_{6.3}$ geometric diameter for spherical dust particles (Hinds, 1999; Rodríguez et al.,
2012) and to $PM_{6.9}$ in the case of dust aspherical particles (Huang et al., 2021). Similarly, $PM_{2.5}$ (aerodynamic diameter) is
equivalent to $PM_{1.6}$ (geometric diameter) for dust.  Using standard relationships between the modal particles used in the
CAM6 (Liu et al., 2016) and the fraction of the particles below 6.9 µm (Seinfeld and Pandis, 2006) (here referred to as
$PM_{6.9}$), a new diagnostic was added to the model, which shows that in regions with substantial coarse particles like dust,
there can be a difference of about 30%, while in most places the differences are less than 5% (Fig. S1). These assumptions
are less true for coarse particles like sea salts, but the differences are small in sea salt dominated regions (Fig. S1).  For this





study we use $PM_{6.9}$ from the model. Note that the inlet size discrimination for $PM_{2.5}$ measurements are also not a step
function and also this might affect the comparisons for $PM_{2.5.}$
For ease of viewing the data in this paper in the densely sampled regions as well as to compare model output to more
representative spatial scales, observational records from different sites were combined into a mean within a grid cell that is
two times the model resolution, or approximately 2° x 2°. This process averages the observations over a spatial scale
appropriate for comparison with the chemistry model (Schutgens et al., 2016). We provide both the climatological annual
average data at each site as well as the averaged data (with the modelled data at doi: 10.5281/zenodo.10459654, Mahowald
et al., 2024).
Notice that we include both urban regions and rural or remote sites into the same dataset. Some of the original meta data did
not include the resolution of the location to better than 0.25 degrees, so that the coordinates of the locations here provided
with the gridded data should not be used for finer resolution studies. Because of the importance and size of megacities,
which cross multiple grid boxes, we include urban and rural air quality data in the same dataset, and previous studies show
the expected differences between urban and rural concentrations and trends (e.g., Hand et al., 2019).
There are multiple sources of uncertainties in the observations used in the model-data comparisons of PM concentrations at
the global model grid scale: errors in the measurements, differences in measurement methods, variability in aerosol
concentrations during events versus background conditions, spatial variability within a model grid box, and interannual
variability. To assess the size of these uncertainties, we look at the normalized standard deviation (defined as the standard
deviation over the mean) due to these factors in the observations for within year, with in a 2°x2° degree grid and for
interannual variability. To evaluate within year and between year variability, we focus on stations that have more than 10
years of data. To evaluate spatial variability within grid boxes, we use grid boxes that have more than 10 stations within
them. Notice that these grid boxes are likely to lie close to cities and fossil fuel source regions, because the measurement
network is more dense there, perhaps exaggerating the importance of spatial variability. In addition, different measurement
methods (dry vs. moist aerosol mass different inlet geometries) complicate the comparisonl of data. We assume here a
measurement method uncertainty of 30% that is on the high side of previous studies (Prank et al., 2016; Burgos et al., 2020;
Hand et al., 2017). Many of the measurements also include an assessment of their uncertainty or of the minimum detected
limit: we use that to assess the average uncertainty of individual measurements (measurement errors).



## 3 Results

### 3.1 AEROMAP observational data set

First, we assessed the amount of data and the number of stations within each ~ 2° x 2° gridded area (Fig. 1). The observational dataset provides coverage predominately over North America and Europe for $PM_{2.5}$ and $PM_{10}$, as noted by previous studies (e.g., Szopa et al., 2021), but in addition we provide here a synthesis of more air quality data in other regions, especially Asia (Fig. 1). This data set comprises most of the individual observations (at daily or higher time periods) of total $PM_{2.5}$ (Fig. 1a, 1e: blue bars) and most of the observing stations (Fig. 1e and blue line). Approximately 15,000 stations and over 20 million observations are included in this compilation as annual averages.

Notice that there are two to three orders of magnitude more daily observations for the total mass (PM) of particles compared to information about the composition of particles (Fig. 1e), which is shown also by contrasting the spatial distribution of measurements between $PM_{2.5}$ and measured amounts of OM (Fig. 1a versus 1b), as well as a large difference between the number of stations measuring the total mass versus the speciated aerosol particles like OM (Fig. 1c versus 1d). While this dataset presents a huge increase in the amount of data available to the aerosol modelling community, still the dominant proportion of the total $PM_{2.5}$ or $PM_{10}$ data are clustered over a few regions, and there is little composition information over most of the globe (Fig. 1).

### 3.2 Uncertainties in spatial aerosol distributions

Our goal in this study was to provide observational constraints on particles that vary spatially over 4-5 orders of magnitude globally (Mahowald et al., 2011). To do that we collect all available datasets, prioritizing long term stations with composition data, but in regions with few measurements, we include only PM data, or data collected during field campaigns, which may last only a month or two. Previous studies have shown that even a 1 day average aerosol measurements, carried out on cruises, can constrain aerosol concentrations within a order of magnitude (1-sigma) for phosphorus in dust, which varies spatially by 4 orders of magnitude (Mahowald et al., 2008). Other studies have highlighted that even for particles that have highly variable sources, such as dust, that only a few months of observations are enough to characterize the mean and standard deviation in most places across the globe (Smith et al., 2017). However, that study highlighted that places where dust events do not occur every year, like near South America, several years are required to characterize the mean (Smith et al., 2017). Thus, the dataset described here cannot do a good job of constraining aerosol concentrations that are due to episodic emission events like wildfires or dust in regions without long term datasets.

Uncertainties in the observations used in the comparisons of aerosols at the global model grid scale come from multiple sources: errors in the measurements, differences in measurement methods, variability in aerosol concentrations during events versus background conditions, spatial variability within a model grid box, and interannual variability, as discussed in Section 2.3. To assess the size of the variability contribution to the uncertainties, we look at the normalized standard deviation



(defined as the standard deviation over the mean) due to these factors in the observations for within year, with grid and
interannual variability. In addition, different measurement methods (dry vs. moist aerosol mass different inlet geometries)
complicate the comparison of data. We assume here a measurement method uncertainty of 30% that is on the high side of
previous studies (Prank et al., 2016; Burgos et al., 2020; Hand et al., 2017). Many of the measurements also include an
assessment of their uncertainty or of the minimum detected limit: we use that to assess the average uncertainty of individual
measurements (measurement errors).
The largest uncertainties are associated with within-year variability (0.45) (Figure 1g). Uncertainty due to combining
different measurement methods (0.3) and from spatial variability within a model grid cell (0.23) are also important (Figure
1g). Both interannual variability (0.12) and measurement errors (0.1) are smaller but important contributions to uncertainty.
The importance of within year variability is consistent with studies showing that in most places, there are a few pollution
events carrying much of the mass, and with otherwise much lower background concentrations (Luo et al., 2003; Fiore et al.,
2022). Obviously, interannual variability is important for secular trends (Gupta et al., 2022; Watson-Parris et al., 2020), but
tends to be much smaller than the 2-4 orders of magnitude of the spatial variability across the globe, and thus can be neglected
for understanding global spatial distributions (Figure 1g).
These sources of uncertainties occur simultaneously and if we sum them assuming orthogonality, we obtain an
normalized uncertainty of 0.6, which was interpreted as meaning that model/data comparisons within a factor or three should
be considered adequate. To ease the visual evaluation of the comparison we show in the following scatter plots both the 1:1
line and the range within a factor of 3. We discuss an example of uncertainties in more detail in Section 3.3. Notice that if we
use the same metric (normalized standard deviation) to evaluate the variability across the climatological concentrations
measured in the observations at different locations (Figure 2a) or across the grid averages in the model we obtain 1.0 and 2.2,
respectively, much larger than the uncertainties (0.6): there is much more variability across different grid boxes (4-5 orders of
magnitude) than in time (up to 50%). As expected, the model contains more spatial variability than the observations, as the
model reports concentrations in very high (North Africa) and very low (Antarctica) aerosol regions where we have no data.
**3.3 PM$_{2.5}$ model-data comparison**
Modelled concentrations of PM$_{2.5}$ are more often compared against observations than for PM$_{10}$ or other size fractions, and
comprise an important portion of the particulate matter associated with human activities. Therefore, we describe first the
observational synthesis and comparison to model results for PM$_{2.5}$. Because the high number of observations in some parts of
the world would make the figures unreadable, the observations are gridded onto an approximately 2°x2° grid for
comparisons with the model (Fig. 2a). As expected, in the model the highest concentrations are over the desert dust regions,
such as North Africa, and over heavily industrialized regions in Asia. For the heavily industrialized regions in Asia, these
high values are consistent with the observations, but the regions in North Africa with the highest modelled values do not
have similar observational validation for high concentration values due to a lack of data (Fig. 2a).



Overall, the model is able to simulate much of the spatial variability in PM$_{2.5}$ over two orders of magnitude (Fig. 2a and 2c),
however there is an overestimate in the PM$_{2.5}$ over India and China (Fig. 2b), which for some observations is outside the x3
uncertainty estimates (Figure 2c and 2d). As an example of the source of the uncertainties discussed in Section 3.2, we
discuss these in more detail.  It seems likely that at least some of these errors are due to an overestimate in the emission
databases, since satellite based remote sensing has shown an overestimate in SO$_2$ over China (Luo et al., 2020).  In addition,
these discrepancies could be due to an error in the aerosol transport modelling or the time period: the observations are more
recent while the assumptions for the emissions are for the year 2010. In addition, notice that once averaged over the 2x2
grids more observations are within a factor of 3, our uncertainty (contrast 2c and 2d).  However, there could also be
methodological and analytical differences due to which group or network did the observations or the exact locations of the
different monitors.  Much of the data in those regions are not usually included in compilations of data, so the fact that
previous model studies have not been able to assess emission datasets in these regions could explain this discrepancy.
Comparison between different observations in some cities (Fig. 3) shows that in these grid boxes there can be very large
differences (~factor of 3) between the annually averaged values reported at nearby stations within 1° distance radially.
Notice that the AirNow measurements (**https://www.airnow.gov/international/us-embassies-and-consulates/** on the US
embassies) tend to be higher than those reported from government air quality networks. The sites compared are in large cities
and thus are likely to have strong local sources and intense gradients in pollutants. For now, we keep in mind this large
difference, but continue to use the observations.  As indicated below, in these regions we do not have measurements of
composition so we do not know which constituents are poorly simulated in our emissions or transport modelling.  More
statistics describing the model data comparisons are shown in Table S4.
The scatterplots show the comparisons of the model to the observations using the gridded data (Fig. 2c) and all original data
(Fig. 2d), and the correlation coefficients are similar (0.73 vs. 0.78 in Fig 2c and Fig 2d, respectively).
Next, we consider the composition of the PM$_{2.5}$ aerosol in the model versus the observations, starting with the aerosol
components in the default version of the model. Sulfate particles tend to be overestimated in the model in North America,
but not over Europe and other regions (Fig. 4a and b).  Previous studies have compared SO$_4^{2-}$ aerosol observations to some
model simulations and have not noted this bias (e.g., Barrie et al., 2001; Aas et al., 2019) but this bias was seen in this model
(Liu et al., 2012; Yang et al., 2018).  BC comparisons suggest the model results are roughly able (r=0.25, within the x3
uncertainty) to capture the spatial dynamics of this aerosol across more than 2 orders of magnitude (Fig. 4c and d).  This is
similar to previous model intercomparisons (Koch et al., 2009; Bond et al., 2004, 2013; Liu et al., 2012, 2016).  Simulations
of OM in the default model suggest that the model is within the uncertainty of most of the data.  Correctly modelling organic
material is very difficult both due to the sparsity of data for comparison, as well as the importance of both primary and
secondary OM in PM (Heald et al., 2010; Kanakidou et al., 2005; Olson et al., 1997; Tsigaridis et al., 2014), and previous
studies with this model have noted an overestimate in comparison with surface observations (Liu et al., 2012).  In our study



we include primary biogenic particles, which are usually not included in model studies (Mahowald et al., 2011, 2008;
Jaenicke, 2005; Heald and Spracklen, 2009; Burrows et al., 2009; Myriokefalitakis et al., 2016), but these are a very small
part of the PM$_{2.5}$ and occur mostly in the coarse fraction (Table 1) and thus are not causing any bias, which must be due to
biomass burning and/or industrial emissions.
As a proxy for sea salts, we use the elemental data of the major component, Na, and although most of the data is within the
uncertainties, the model tends to be too high at low Na and too low at high Na in North America, where much of the data are
available (Fig. 4g and h), which has been seen previously with this model (Liu et al., 2012). Notice that we do not include
industrial emissions of Na as they have not been spatially estimated. As a proxy for dust, we use Al amounts (Fig. 4i and j),
which globally and over dust regions are dominated by dust, although there are few observational datasets in high dust
regions. The comparisons suggest the model is able to simulate dust across 4 orders of magnitude, similar to previous studies
(Liu et al., 2012; Albani et al., 2014a; Li et al., 2022b; Huneeus et al., 2011) although there is a tendency for a high bias in
the models over low dust regions and a low bias in high dust regions, similar to sea salts (Fig. 4i and 4.j).
Next, we consider the nitrogen aerosol ammonium nitrate that requires complicated gas-aerosol phase equilibrium to
correctly simulate (e.g., Bauer et al., 2007; Thornhill et al., 2021; Adams et al., 2001; Regayre et al., 2018; Seinfeld and
Pandis, 2006). To summarize these complicated interactions, because $SO_4^{2-}$ is a stronger acid than $NO_3^-$ in the atmosphere,
the basic $NH_4^+$ is preferentially found with $SO_4^{2-}$. Thus $NO_3^-$ particles will only form if there is sufficient $NH_4^+$ available,
therefore the ratio of $NO_3^-$ to total $NH_4^+$ can vary. As described in the methods, to include these particles we used
simulations from a different version of the same model which include chemistry (Vira et al., 2022), and a more process-
based source of ammonia (Vira et al., 2020) since the default CESM2 version used here for most particles does not include
chemistry. Note that even in the chemistry version of the model for CESM2 the complicated gas-aerosol phase equilibrium is
not included, which causes errors in the simulation of the amounts of nitrogen aerosol (e.g., Bauer et al., 2007; Thornhill et
al., 2021; Adams et al., 2001; Regayre et al., 2018; Nenes et al., 2021). Thus while the NH$_3$ agricultural emission scheme
used in this model is state-of-the-art, the lack of an adequate gas-aerosol phase separation may lead to biases as discussed in
Vira et al. (2022). In addition, recent studies have suggested that emissions of NH4 from vehicles should be 1.8x higher
than previously estimated (Toro et al., 2024), highlighting the difficulty of adequate emission datasets for nitrogenous
aerosol precursors. $NO_3^-$ particles compared against available observations show that over 2 orders of magnitude, the model
results are able to simulate the spatial variability, with most of the data within the uncertainties (Fig. 4k and l). Note that
here, we have multiplied the simulations by a factor 0.5 in order to achieve a better mean comparison, as indicated by Vira et
al. (2022). In addition, $NH_4^+$ results show the importance of $NH_4^+$ over agricultural regions especially (e.g., Vira et al.,
2022), and that the $NH_4^+$ in the simulation used here compares well to available observations by being within the
uncertainties at most observational sites (Fig. 4m and n; Vira et al., 2022).





**3.4 PM₁₀ model-data comparison**

PM₁₀ was the first size selective standard for particulate air quality until more studies showed that smaller particles (PM₂.₅ or PM₁) were more relevant for health impacts and PM₂.₅ standards were added (e.g., https://www.epa.gov/pm-pollution/timeline-particulate-matter-pm-national-ambient-air-quality-standards-naaqs, accessed October 4, 2023). However, there are still many PM₁₀ measurements routinely made (Fig 1d; Fig. 5a). As discussed in the methods, what is described as measurements of PM₁₀ (aerodynamic diameter) is probably closer to PM₆.₉ (geometric diameter) as simulated in models (Huang et al., 2021), so here we use the PM₆.₉ fraction as calculated in the model to compare to PM₁₀ observations (Fig. S1 shows the fraction of PM₁₀ that is PM₆.₉. This distinction is only important in regions with substantial coarse mode emissions like desert dust source regions. For marine coarse aerosols like sea salt, the distinction between geometric and aerodynamic diameter may be smaller.). The model is able to simulate PM₁₀ concentrations across 2 orders of magnitude with some skill, as most of the data is within the uncertainties (Fig. 5a, c and d), although the region of East Asia, especially China and India are overestimated in the model similar to the PM₂.₅ (Fig. 3a, and b). Gridding the data before comparing to the model results in a similar correlation across space as including all data (Fig 5b vs. c). More statistical comparisons are shown in Table S5.

There are fewer comparisons with PM₁₀ composition data available in the literature: usually only sea salts and dust are compared to observations that include the coarse mode (Gong et al., 2003; Ginoux et al., 2001; Albani et al., 2014b; Mahowald et al., 2006). Comparisons for $SO_4^{2-}$ suggest that the model tends to over predict PM₁₀ values in some locations, as many observations are too high and outside the uncertainty bounds (Fig. 6a and b.). For BC, the PM₁₀ simulation captures the range of values, with most of the data within the uncertainty bounds (Fig. 6c and d in contrast to a and b). Unlike many studies we include BC in the PM₁₀ mode, since observations show that there are some contribution of BC to PM₁₀ (compare Fig. 6c versus 4c). The model simulations for OM include primary biogenic particles and the limited available observations do not support larger sources of OM in the PM₁₀, than included here (as suggested in e.g., Jaenicke, 2005): indeed the model is overestimating the OM in PM₁₀ at many stations especially in North America. Similarly, the limited Na (indicating sea salt) data suggest the model in some places may overestimate Na even over continents outside the error bound (Fig. 6g and h), as discussed in the PM₂.₅ section, as was seen previously (Liu et al., 2012). Comparisons with Al (proxy for dust) show that the variability is well simulated, but the model overpredicts the concentrations. Dust models are compared against aerosol optical depth at stations and using global averages of deposition and surface concentrations and it is currently not possible to simulate all of these at the same time, consistent with previous studies with this model (Li et al., 2022b; Kok et al., 2014b; Albani et al., 2014a; Matsui and Mahowald, 2017; Zhao et al., 2022), and indeed across most dust models (Huneeus et al., 2011).





For the particulate $NO_3^-$, similar to the $PM_{2.5}$ size, the particles were multiplied by 0.5 to better match the observations
following Vira et al. (2022) (Fig 6k and l). The model simulations suggest too high values in high $NO_3^-$ areas, and too low in
low $NO_3^-$ regions (Fig. 6k and l). $NH_4^+$ shows a slightly better comparison to the limited available data (Fig. 6m and n) as
seen in Vira et al. (2022). As discussed earlier, the model does not include other forms of nitrate aerosols which may be
important, such as the reaction of nitrate with dust aerosols (Wolff, 1984; Dentener et al., 1996; Xu and Penner, 2012;

**3.5 Data and model coverage**

The compilation shown here is the most comprehensive currently available for describing the spatial variability of the total
mass and composition of in situ particulate concentration data, and yet it highlights the lack of sufficient data to constrain the
current distribution of particles and their composition (Fig. 7a and b). Only 3% of the grid boxes (2°x2°) have $PM_{2.5}$ data
(about 10% of land grid boxes), and only 0.3% has sufficient data to constrain most of the composition (defined as having
90% of the variables considered here: total mass, $SO_4^{2-}$, BC, OM, Na or Cl, Al or dust, $NO_3^-$ and $NH_4^+$). There are even less
data available to characterize $PM_{10}$, which is less important for air quality and aerosol-cloud interactions but more important
for aerosol-biogeochemistry interactions and long wave interactions (Mahowald et al., 2011; Li et al., 2022a; Lim et al.,
2012; Kanakidou et al., 2018). Because of the high spatial and temporal variability and the lack of satellite or other remote
sensing data to characterize the type of aerosol, this lack of data is a severe handicap in constraining aerosol radiative forcing
uncertainties and other impacts of particles in the climate system.
In this simulation, we included several new aerosol sources and types that are not in the default model to investigate their
importance. For the CESM this simulation includes agricultural dust, nitrogen particles and several other sources (see Table
1). As shown in Fig. 8, the default particles are the dominant particles over most of the planet, but in many regions for both
$PM_{2.5}$ and $PM_{10}$, the default aerosol scheme includes less than 30% of the aerosol particles (Fig. 8a and c), with substantial
contributions from the new added particles (Fig. 8b and d), especially nitrogen particles and agricultural dust. Many Earth
system or climate models such as the CESM2 do not include nitrogen particles ($NO_3^-$ and $NH_4^+$), because of the substantial
complexity and computation load of chemistry and gas-aerosol equilibrium (Bauer et al., 2007; Thornhill et al., 2021; Adams
et al., 2001; Regayre et al., 2018)). Previous studies have highlighted the importance of nitrogen particles for climate, air
quality and ecosystem impacts (e.g., Adams et al., 2001; Bauer et al., 2007, 2016; Kanakidou et al., 2016; Baker et al.,
2021). Changes in nitrogen aerosol emissions are likely to follow different future trajectories than $SO_4^{2-}$, BC or OC, whose
anthropogenic sources are mostly fossil fuel derived and should decrease in the future as renewable energy resources expand
(Gidden et al., 2019). Ammonia has substantial sources from agriculture, which will likely to stay constant or expand
(Gidden et al., 2019; Klimont et al., 2017; Bauer et al., 2016). This suggests there could be a substantial bias in both
historical and future aerosol forcings due to the lack of inclusion of these important sources (e.g., Bauer et al., 2007;
Thornhill et al., 2021; Adams et al., 2001; Regayre et al., 2018).



## 4. Conclusions

In this study, we present a new aerosol compilation (AERO-MAP) designed to evaluate the spatial variability of particulate matter in Earth system and air quality models. This climatologically averaged dataset includes both total mass and composition, where available, including 15,000 station datasets and over 10 million daily to weekly averaged measurements. Spatial variability represents the largest source of variability in aerosols (Figure 1f and Section 3.2), and thus the most important to simulate accurately in models, especially as some climate effects are strongly non-linear, and knowing small concentrations ($1 \; \mu g/m^3$) versus very small concentrations ($0.1 \; \mu g/m^3$) is important (Carslaw et al., 2013). Here we expand beyond the usual limited coverage of only North America and Europe to present a more global data view for both $PM_{2.5}$ and $PM_{10}$ (Fig. 1). Unfortunately, there are still very limited data characterizing both the surface concentration, size and composition of aerosol particles (Fig. 7). While satellite remote sensing can indicate the total atmospheric loading during cloud free conditions, it cannot yet provide substantial information about the size or composition of particles (Kahn et al., 2005; Tanré et al., 1997; Remer et al., 2005). Surface based remote sensing may provide more information about size and absorption properties (Holben et al., 1991; Dubovik et al., 2002; Schuster et al., 2016; Gonçalves Ageitos et al., 2023; Obiso et al., 2023), but single scattering albedo, for example, is only available under very high (>0.4 AOD) aerosol loading conditions, and thus is not available most of the time and space (Dubovik et al, 2002). Knowing the size and the composition of aerosols is key to their impacts on air quality and climate (Mahowald et al., 2011). Knowing what particulates are dominant in a region is required, as fossil fuel derived aerosols will be reduced, while agriculturally based aerosols may well increase (Gidden et al., 2019). We also present a method that is generalizable to other models to use this dataset to evaluate both mass and composition for intercomparison projects and improvements in air quality and Earth system models.

This study has highlighted the value of surface concentration data, but represents only the climatological mean values showing the spatial variability, while there is also information in the temporal variability of the PM. A recent, independent and complementary effort collects all atmospheric composition data (not just aerosols) from many networks into one easy to use framework called GHOST (Globally harmonised dataset of surface atmospheric composition measurements; Bowdalo et al., 2024). The approach used in GHOST includes presenting the data in netcdf format, at the original resolution, with meta data about measurement type, etc. included, and is an important step forward (Bowdalo et al., 2024). At this point GHOST only includes a subset of the data available in this study: we hope that the GHOST effort can be expanded to include more spatial variability and be maintained into the future.

This study also highlights the importance of including all aerosol components into the models, and shows that in the CESM2, in many places there is between 10-60% of the particulate mass missing, largely due to lack of the nitrogenous particles (Paulot et al., 2016; Adams et al., 1999; Thornhill et al., 2020) and the poorly understood agricultural dust particles (e.g., Ginoux et al., 2012). Because these particles are largely driven by agricultural sources and not fossil fuels, their





concentrations will be hardly affected by the transition to renewable energy and may increase if agricultural production
expands with population. Therefore, these aerosol particles represent important air quality and climate impacts that should
be represented more accurately in future studies.
**Data availability:** The data compiled here is available as a csv table with citations as a supplemental data. This same file is
available as well as gridded datasets with the modelled data in netcdf format at https://zenodo.org/records/10459654,
**10.5281/zenodo.11391232** Mahowald et al., 2024. Additional underlying datasets available by request to
mahowald@cornell.edu.
**Code availability:** The model used here is a version of the Community Earth System Model, and the modifications and input
files to that code are available at https://zenodo.org/records/10459654, Mahowald et al., 2024.
**Author contributions:** NMM designed and oversaw the implementation of the approach with the advice of HL, CW, RVM
and JL, and wrote the first draft of the manuscript. JV, PH, LLi, ZK, CD, SR, TB and DH assisted in the version of the
model and emission datasets used. EA, DM, HM and LLu authors assisted in the compilation and conversion of the data,
CH, ZKl contributed emission datasets, XL and XZ contributed model code, MGA, CA, AA, PA,AB, FB, SB, GC, SC, YC,
PC, DC, CC, ED, GD, KE, CG-L, CG, DG, YGR, HH, RH, CH, BH, PH, CH, MK, ZKe, KK, FL XL, RL, RL, WM, BM,
RM, NM, YM-G, AP, JP, SR, PS, DV, BW authors contributed data. All authors edited the manuscript.
**Competing interests:** The authors declare that the only conflict of interest is that Maria Kanakidou, Xiaohong Liu, Willy
Maenhaut, and Sergio Rodriguez are editors at Atmospheric Chemistry and Physics.
**Acknowledgments**
NMM and LL would like to acknowledge support from DOE (DE-SC0021302 ), as well as from Paul Ekhart (EBAS), the
many freely available air quality websites acknowledged in the paper: EBAS (https://ebas.nilu.no/)--including data affiliated
with ACTRIS (Aerosol Clouds and Trace gas Research Infrastructure), EMEP (European Monitoring and Evaluation
Programme), GAW-WDCA (Global Atmosphere Watch-World Data Centre for Aerosols), EANET Acid Deposition
Monitoring Network in (East Asian)--All Indian Air Quality Management data (https://app.cpcbccr.com/ccr/#/caaqm-
dashboard-all/caaqm-landing/data), Australian National Air Pollution Monitoring Database (https://osf.io/jxd98/), South
African Air Quality Information System (https://saaqis.environment.gov.za/), Mexico City Air quality data



( http://www.aire.cdmx.gob.mx/default.php?opc=%27aKBh%27), Chile ( Sistema de Informacion Nacional de Calidad del
Aire--https://sinca.mma.gob.cl/index.php/), Japan's NIES (National Institute for Environmental studies-
https://tenbou.nies.go.jp/download/), Turkey Air Quality Monitoring Network, Israel Air Quality Monitoring website, US
EPA CASNET and IMPROVE, US AIRNOW, New Zealand Stats now website, Chilean
(https://www.stats.govt.nz/indicators), Chinese Air Quality data collected together (https://osf.io/jxd98/) and Canadian
National Air Quality Surveillence (https://data.ec.gc.ca/data/air/monitor/national-air-pollution-surveillance-naps-
program/Data-Donnees).   FB and FL would like to acknowledge support from the Ministerio del Medio Ambiente de Chile
(https://mma.gob.cl) and Fondecyt 1231682. SC is grateful for financial support from the Texas Air Research Center and the
Texas Commission on Environmental Quality. PA acknowledges funding from Fundação de Amparo à Pesquisa do Estado
de São Paulo (FAPESP), grants number 2017-17047-0 and 2023/04358-9. RVM acknowledges funding from NSF Grant
2020673. MK and NM acknowledge support by Greece and the European Union (European Regional Development Fund)
via the project PANhellenic infrastructure for Atmospheric Composition and climatE chAnge (PANACEA, MIS 5021516).
CGL and BM acknowledge support of CNRS, IRD and ACTRIS-France to the International Network to study Atmospheric
Deposition and Atmospheric chemistry in AFrica programe (INDAAF).   HM acknowledges support by the MEXT/JSPS
KAKENHI Grant Numbers JP19H05699, JP19KK0265, JP20H00196, JP20H00638, JP22H03722, JP22F22092,
JP23H00515, JP23H00523, and JP23K18519, the MEXT Arctic Challenge for Sustainability phase II (ArCS II;
JPMXD1420318865) project, and by the Environment Research and Technology Development Fund 2–2301
(JPMEERF20232001) of the Environmental Restoration and Conservation Agency.  RLM acknowledges support from the
NASA Modeling, Analysis and Prediction Program.  We acknowledge contributions from Sagar Rathod, Tami Bond, Giles
Bergametti, Javier Miranda Martin del Campo, and Xavier Querol. The support to CESAM by FCT/MCTES
(UIDP/50017/2020+UIDB/50017/2020+ LA/P/0094/2020) is also acknowledged.

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

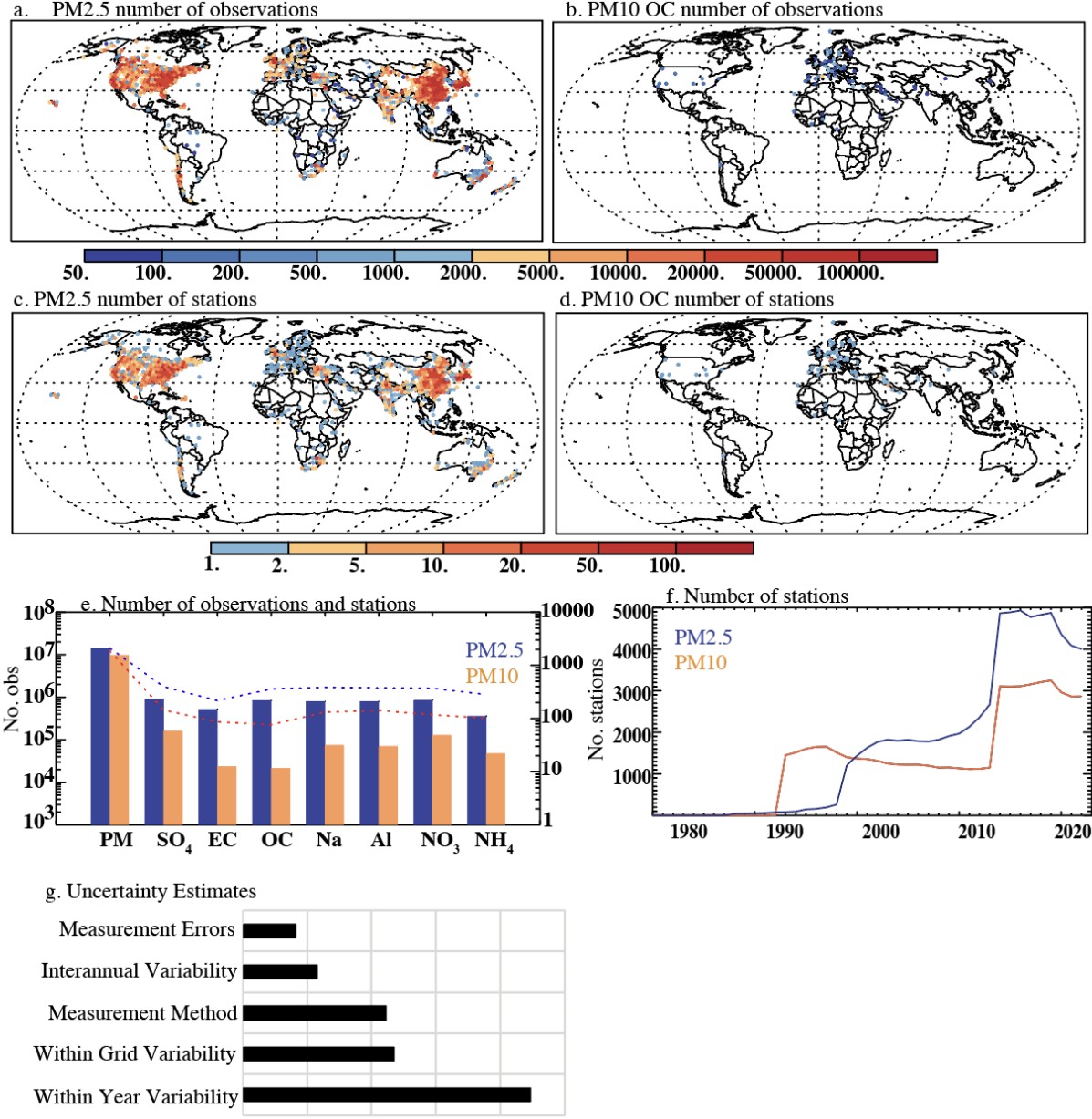

**Figure 1:** Distribution of observations in the data base, showing the number of observations of PM$_{2.5}$ (a) and PM$_{10}$ organic carbon (OC) (b) (with the colors indicating different numbers using the top color bar), as well as the number of stations within each 2x2 grid locations for PM$_{2.5}$ (c) and PM$_{10}$ OC (d) (using the second color bar), showing that there is much more PM$_{2.5}$ or PM$_{10}$ data, in contrast to speciated data. e) The number of observations (bars) for total particulate matter (PM) or speciated data is summarized for the PM$_{2.5}$ (blue) and PM$_{10}$ (orange) fraction using the left-hand side y-axis. The number of stations included in the study is shown as a dotted line (e)



and uses the right-hand size y-axis.  f) The number of stations of PM$_{2.5}$ (blue) and PM$_{10}$ (orange) for each year is
shown. g) Normalized (1 standard deviation over the mean) observational uncertainty for PM$_{2.5}$ from
measurement errors, interannual variability, measurement method, within grid variability and within year
variability at the same station.  Interannual variability and within grid uncertainty are defined as the normalized
standard deviation in the variabilityfor stations that have more than 10 years of data.  Within grid variability is the
normalized standard deviation of 2x2 grid cells that have more than 10 stations.  Measurement errors are the
normalized standard deviation of the reported measurement errors for PM$_{2.5}$. Measurement method error derives
from differences between different measurement methods (e.g., Prank et al., 2016; Burgos et al., 2020; Hand et
al., 2017).  The stations included derive from the following sources (see supplemental dataset for more details):
Alastuey et al., 2016; Almeida et al., 2005; Amato et al., 2016; Andreae et al., 2002; Arimoto et al., 2003; Artaxo
et al., 2002; Barkley et al., 2019; Barraza et al., 2017; Bergametti et al., 1989; Bouet et al., 2019; Bozlaker et al.,
2013; Chen et al., 2006; Chuang et al., 2005; Cipoli et al., 2023; Cohen et al., 2004; da Silva et al., 2008;
Dongarrà et al., 2007, 2010; Engelbrecht et al., 2009; Formenti et al., 2003; Fuzzi et al., 2007; Hand et al., 2017;
Heimburger et al., 2012;     Herut and Krom, 1996; Herut et al., 2001; Hsu et al., 2016; Hueglin et al., 2005; Furu
et al., 2022, 2015; Gianini et al., 2012a, b; Kalivitis et al., 2007; Kaly et al., 2015; Kubilay et al., 2000; Kyllönen
et al., 2020; Laing et al., 2014b, a; Lucarelli et al., 2014,     2019; Mackey et al., 2013; Maenhaut et al., 1996c, a,
b, 1997a, b,     1999, 2000a, 2000b, 2002a, b, 2005    , 2008, 2011; Maenhaut and Cafmeyer, 1998;     Malm et
al., 2007; Marticorena et al., 2010; Mihalopoulos et al., 1997; Mirante et al., 2010    , 2013; Mkoma, 2008;
Mkoma et al, 2009; Morera-Gómez et al., 2018, 2019; Nava et al., 2015, 2020; Nyanganyura et al., 2007;
Oliveira, 2009; Oliveira et al.,     2010; Pérez et al., 2008; Pio et al., 2022    ; Prospero et al., 1989, 2012, 2020;
Prospero, 1996, 1999; Putaud et al., 2004, 2010; Rodríguez et al., 2011, 2015; Salma et al., 1997; Savoie et al.,
1993; Silva et al., 2010; Smichowski et al., 2004; Swap et al., 1992; Tørseth et al., 2012; Uematsu et al., 1983;
Vanderzalm et al., 2003; Virkkula et al., 1999; Xiao et al., 2014; Zihan and Losno, 2016. Data from several
online networks are also included ( e.g., https://www.airnow.gov/international/us-embassies-and-consulates/,
https://quotsoft.net/air/, https://app.cpcbccr.com/ccr/#/caaqm-dashboard-all/caaqm-landing/data,
https://sinca.mma.gob.cl/index.php/. https://tenbou.nies.go.jp/download/).  See the supplemental data set for more
details and the doi links for the datasets.



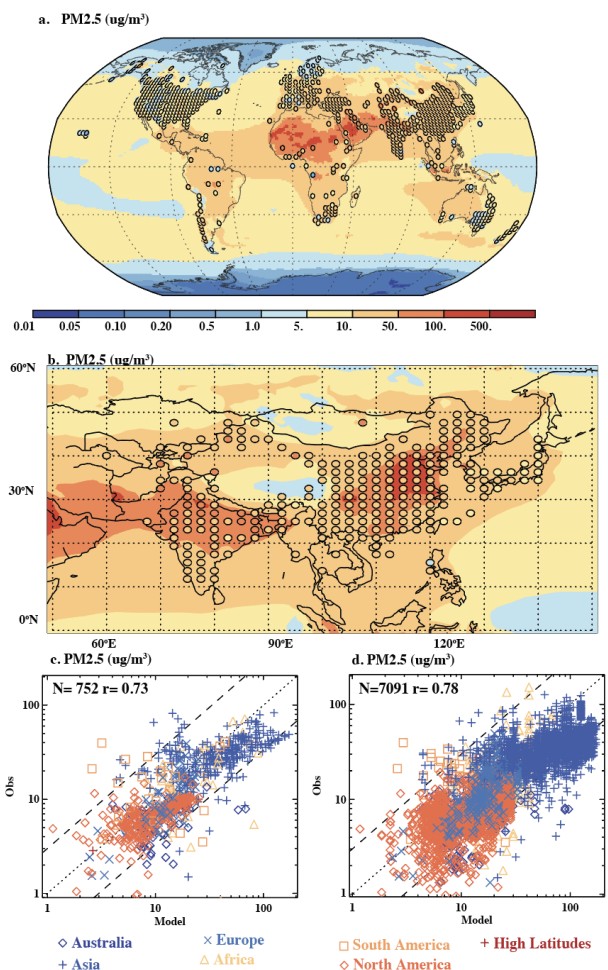

**Figure 2:** Model results and gridded observations for PM$_{2.5}$ in µg/m$^3$ spatially mapped globally (a) and focused on just East Asia (b) where the model is plotted as the background and the observations are circles with the colors indicating the amount of PM$_{2.5}$ using the same scale. A comparison of the model (x-axis) to the observations (y-axis) is shown for the gridded data (c) and including all stations (d). In the scatter plots, the colors and symbols indicate the regions, the dotted line is the 1:1 line and the dashed lines are the factor of 3 uncertainty estimates. More statistics are shown in Table S4, and the model plotted alone is available in Figure S2.



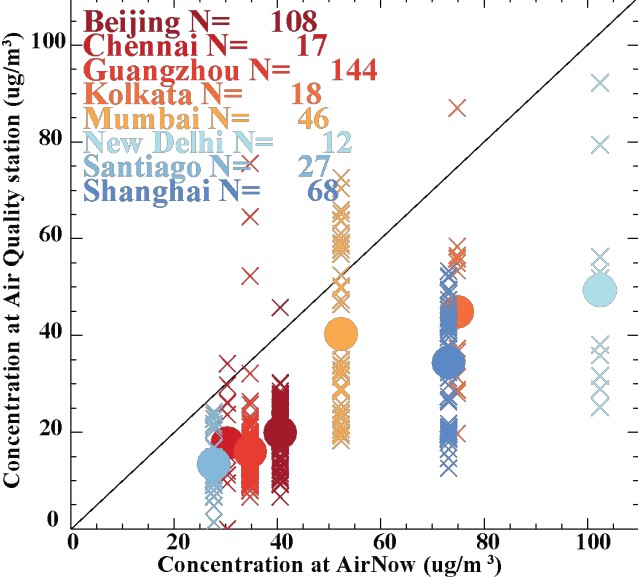

**Figure 3**:    Comparison of PM$_{2.5}$ observations from the US Embassy's AirNow network

(**https://www.airnow.gov/international/us-embassies-and-consulates/**) versus observations from the Chinese

air quality network (downloaded from **https://quotsoft.net/air/**) (Beijing 39.9N 116.4E, Guangzhou 23N 113E,

Shanghai 31N 121E) and the Indian (Chennai 13N 80E, Kolkata 23N 88E, New Delhi 27N 77E) network

(**https://app.cpcbccr.com/ccr/#/caaqm-dashboard-all/caaqm-landing/data**); and observations (Barraza et al.,

2017) from Santiago, Chile (23.7S 70.4W) against the Chilean air quality network

(https://sinca.mma.gob.cl/index.php/).  The numbers after each city name are the number of stations found within

1° distance of the AirNow (or Chile observations) station.







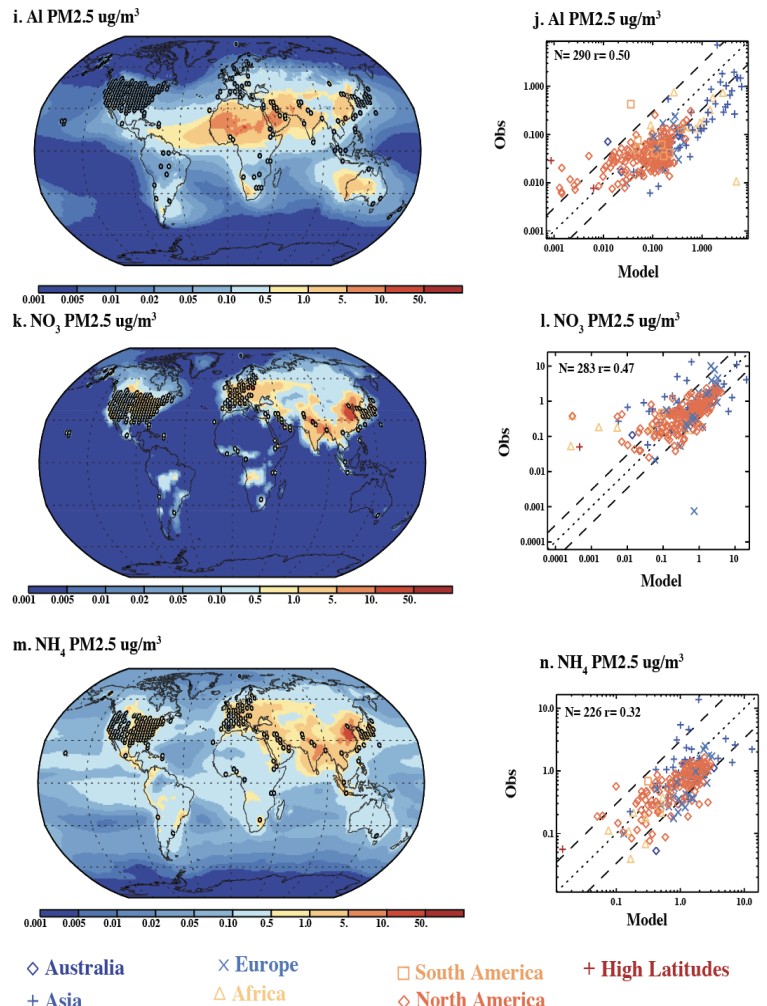

**Figure 4:** Model results and gridded observations for different types of PM$_{2.5}$ in µg/m$^3$ spatially mapped globally where the model is plotted as the background and the observations are circles with the colors indicating the amount PM$_{2.5}$ using the same scale for (a) SO$_4^{2-}$, (c) BC (black carbon), (e) OM (organic material=1.8 times organic carbon (OC)), (g) Na, (i) Al, (k) NO$_3^-$, (m) NH$_4^+$. A scatter plot comparison of the model (x-axis) to the observations (y-axis) is shown for the gridded observational data for for (b) SO$_4^{2-}$, (d) BC (f) OM, (h) Na, (j) Al, (l) NO$_3^-$, (n) NH$_4^+$. In the scatter plots, the colors and symbols indicate the regions, the dotted line is the 1:1 line and the dashed lines are the factor of 3 uncertainty estimates. More statistics are shown in Table S4, and the model plotted alone is available in Figure S2.



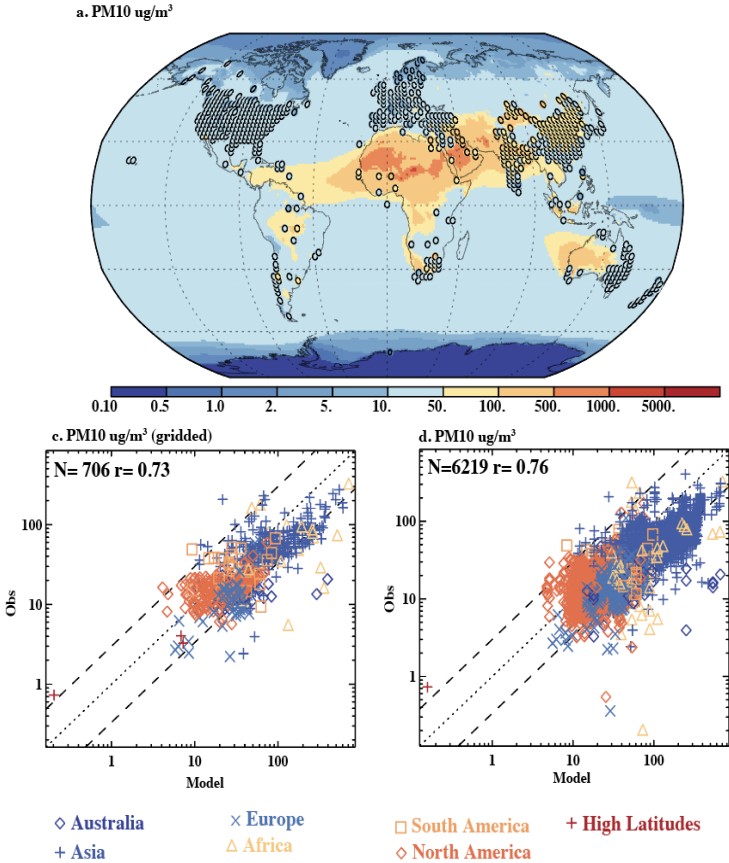

**Figure 5:** Model results and gridded observations for $PM_{10}$ in µg/m$^3$ spatially mapped globally (a). A comparison of the model (x-axis) to the obsevations (y-axis) is shown for the gridded data (b) and including all stations (c). In the scatter plots, the colors and symbols indicate the regions, the dotted line is the 1:1 line and dashed lines are the factor of 3 uncertainty estimates. More statistics are shown in Table S5, and the model plotted alone is available in Figure S3.



**a. SO₄ PM10 ug/m³**

**b. SO₄ PM2.5 ug/m³**

**c. BC PM10 ug/m³**

**c. BC PM2.5 ug/m³**

**e. OM PM10 ug/m³**

**f. OM PM10 ug/m³**

**g. Na PM10 ug/m³**

**h. Na PM10 ug/m³**

◇ Australia    × Europe    □ South America    + High Latitudes
+ Asia    △ Africa    ◇ North America



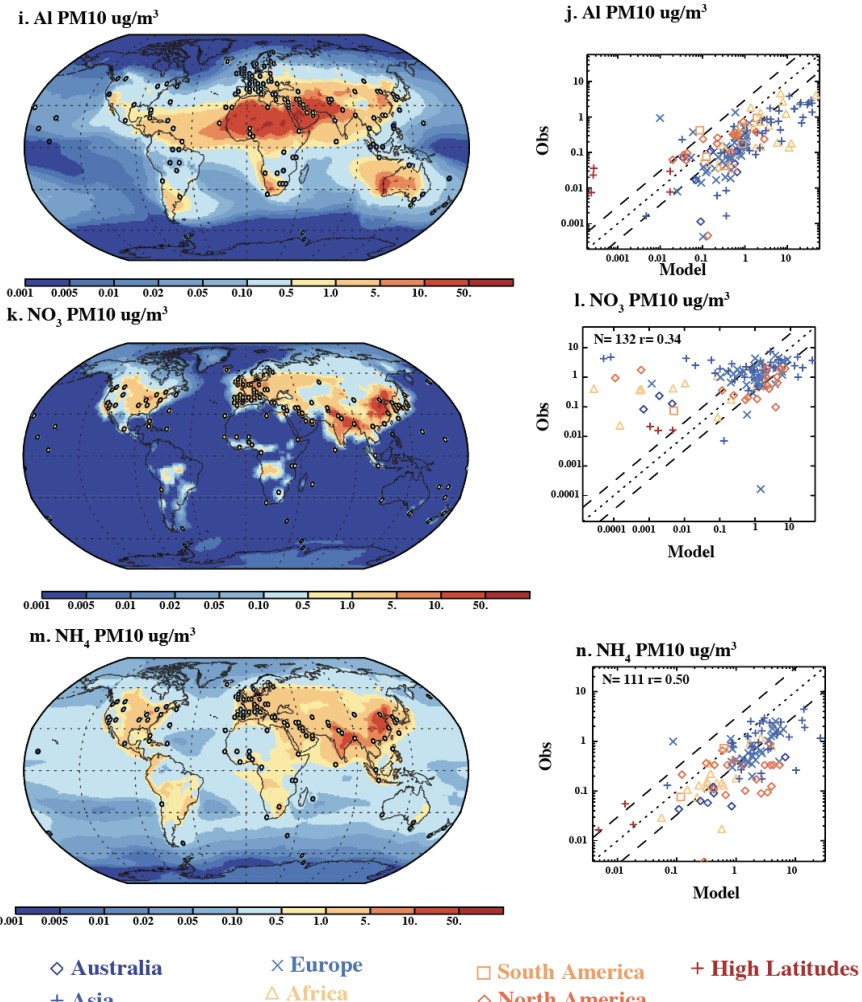

**Figure 6:** Model results and gridded observations for different types of $PM_{10}$ in µg/m³ spatially mapped globally where the model is plotted as the background and the observations are circles with the colors indicating the amount $PM_{10}$ using the same scale for (a) $SO_4^{+2}$, (c) BC (black carbon), (e) OM (organic material=1.8 times organic carbon (OC)), (g) Na, (i) Al, (k) $NO_3^-$, (m) $NH_4^+$. A scatter plot comparison of the model (x-axis) to the observations (y-axis) is shown for the gridded observational data for (b) $SO_4^2$, (d) BC (f) OM, (h) Na, (j) Al, (l) $NO_3^-$, (n) $NH_4^+$. In the scatter plots, the colors and symbols indicate the regions, the dotted line is the 1:1 line and the dashed lines are the factor of 3 uncertainty estimates. More statistics are shown in Table S5, and the model plotted alone is available in Figure S3.



### a. PM2.5 coverage (%)

### b. PM10 coverage (%)

**Figure 7:** Observational coverage (%) for gridded observations, showing within each grid box (2x2) the % of the constituents that are measured assuming that PM, $SO_4^{2-}$, BC, OM, Na, Al, $NO_3^-$, and $NH_4^+$ are required to constrain the PM distribution for (a) PM$_{2.5}$ and (b) PM$_{10}$.



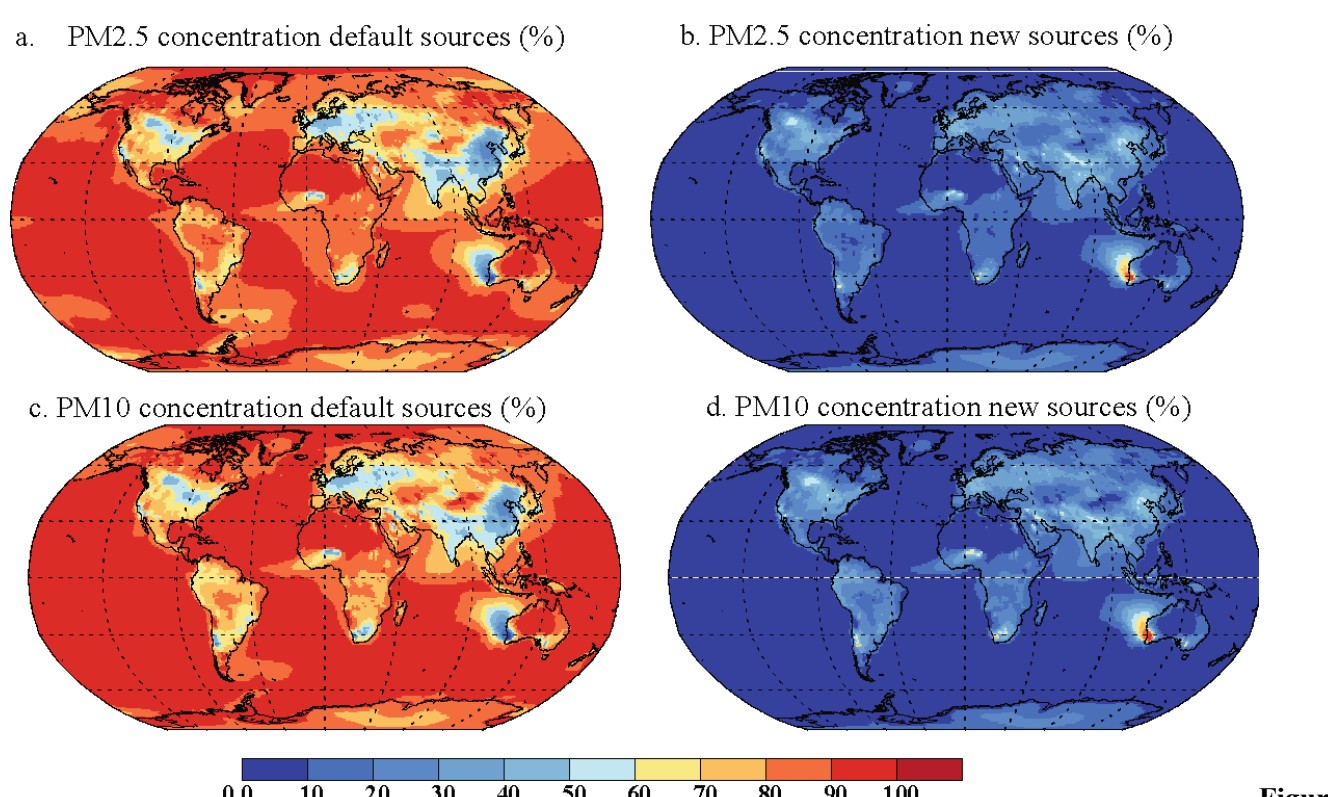

**Figure 8:** Modelled estimates of what percent of the surface concentration of $PM_{2.5}$ is considered in the default CAM (a) or is new in this study (b). Similarly $PM_{10}$ is shown for the default model (c) and new sources in this study (d).





Table 1: Global Aerosol Budgets

Global deposition (Tg/year), percentage of aerosol that is PM$_{2.5}$, and globally and annually averaged surface concentration

($\mu$g/m$^3$) and aerosol optical depth for each of the sources used in the model.  An asterisk indicates that there are additions to

the model from the default CAM6.

| | PM$_{10}$ | PM$_{2.5}$ | | |
| | Deposition (Tg/year) | % | Conc ($\mu$g/m$^3$) | AOD (unitless) |
|---|---|---|---|---|
| Sulfate | 121 | 100 | 2.1 | 0.018 |
| Black carbon | 10 | 100 | 0.5 | 0.009 |
| Primary organic aerosol | 34 | 100 | 1.6 | 0.008 |
| Secondary organic aerosol | 37 | 100 | 1.0 | 0.007 |
| Sea salts | 2520 | 3 | 13.0 | 0.045 |
| Dust | 2870 | 1 | 19.4 | 0.030 |
| NH$_4$NO$_3$* | 20 | 100 | 0.4 | 0.013 |
| Agricultural Dust* | 585 | 1 | 3.7 | 0.006 |
| Road* | 0.43 | 79 | 0.02 | 0.0000 |
| Coarse organic carbon* | 4 | 0.0 | 0.04 | 0.0000 |
| Coarse black carbon* | 0.35 | 0.0 | 0.00 | 0.0000 |
| Fine and coarse inorganic industrial matter * | 56 | 46 | 1.2 | 0.0018 |





| | | | | |
|---|---|---|---|---|
| Bacteria and Fungi spores from land* | 4 | 0 | 0.04 | 0.0000 |
| Other primary biogenic particles from land* | 54 | 3 | 0.4 | 0.0005 |
| Marine organic aerosols | 44 | 99 | 0.6 | 0.0008 |

1495

1496

1497