# Peer review of "AERO-MAP: A data compilation and modelling approach to"

_EGUsphere, 2024_

## Author Response (AR1)

Responses to reviewers
We would like to thank the reviewers for their thoughtful reviews and respond point by point to their reviews.  All responses to reviews are in black and the reviews are in red.

Reviewer #1
This work presents 1) compilation of the available surface PM2.5 and PM10 observations and their composition around the globe with spatially gridded at 2.5 deg of longitude and ~2 deg of latitudes and temporally averaged for any length of measurement duration, 2) presentation the model evaluation of the simulated PM with the gridded climatology dataset as a methodology for model-data comparison, and 3) identification of the measurement and modeling gaps.
1. Data compilation:
Although I commend the team who obviously has put large amount of effort to collect extensive observational datasets (~15,000 stations and over 20 million observations, per section 3.1) and put them into a common format, such compilation at the present format (coarse spatial gridded resolution, no time resolution) provides little usefulness for any type of air quality studies and rather limited help (surface concentration only, no interannual/seasonal variability) for climate model evaluation.

Thank you for your commendation of the large amount of work in creating this compilation.

There is a basic misunderstanding in the dataset we have provided.  The most basic dataset is the climatological annual average at each station, which is contained in the supplemental data set contained with the manuscript as well as in the zenodo archive.   It is unclear from the review exactly what data the reviewer would like to see, but we try to guess what could be most useful.  We also modify the text to make it more clear what is being provided.

We provide the climatological average, a decadal average (2010-2019) and other statistics for each station, with citation in both the Supplemental dataset 1 with this paper, as well as a netcdf file in the zenodo archive.  In addition, we supply annual averages for each year that data is available and the climatological monthly mean.

As an example, we include the following in Section 2.1 (lines 217-245):

"For this paper, we focus on the climatological means for 1986-2023 and decadal means for 2010-2019.  The first period is chosen as the full duration of the individual data sets comprising the compilation are available; the second is chosen to recognize decadal variations in anthropogenic emission within the longer period and isolate a particular decade when data is most plentiful.  In addition, annual means for each year the data is available is also calculated, as well as the climatological monthly means.  The temporal means are calculated for all values at each station that are above the detection limit and reported here. At some stations or times, concentrations can be below the detection limit, and excluding these data or time periods could bias our average values.  We focus on the stations that have more than 50% of the data above the detection limit, and exclude other sites. For those included stations, if the values were reported as below the detection limit, we include in the average one-third of the minimum detection limit. The reported detection limits should bound the upper limit of aerosol mass and allow us to

include sites, whose observations were otherwise too low to include, while reducing the potential biasing of our compilation towards higher values (Supplemental dataset 1).

Our goal is to create easy-to-use datasets for model-data comparisons. Included in this dataset are several files with different levels of description and analysis. One file provides traceability information, including a detailed citation, type and number of measurements included, as well as time period, climatological and decadal (2010-2019) means and standard deviations for each time period (Supplemental dataset 1). For each station dataset included in the database, there will be one line in this file. This means that for some stations (for example K-puszta), there are multiple lines in the supplemental file indicating the two different time periods where measurements were made as well as the two sizes that are measured during each time period. For each station dataset, there are latitude, longitudes, annual mean values, number of observations, year extent of the observations, standard deviations, etc, as well as the citation and where to obtain the data. There are also several netcdf files available at https://zenodo.org/records/11391232 for this dataset. The most useful is likely to be the Allobservation.AEROMAP.nc file, which contains the same quantitative data for each station dataset as the supplement, except that the data is processed to be only $PM_{2.5}$ and $PM_{10}$ (with some TSP data in places with little data, as discussed above). That means $PM_{2.5}$ and coarse aerosol mass are added together if the station datasets are collocated to create a PM10 dataset (e.g., see Table 1). In addition, this file contains climatological monthly means, and annual means for each year for each station dataset, so that temporal information is also easily available. Another file includes the climatological mean observations averaged up to a 2°×2° grid that is used for plotting the figures shown in the paper (Allobservation.AEROMAP.2x2.nc). As indicated in the data availability, only the time-means are available and the underlying data for some datasets cannot be openly published, but please contact the authors (identified by the citation) if other time periods are desired.

,,

The authors justify the compilation of the observational data into coarsely gridded climatological data as to characterize the spatial variability of aerosols. I am very puzzled with that purpose. Considering the large heterogeneity of the sources of aerosol and precursor gases and short lifetime of aerosols, such spatial differences across different aerosol regimes and environmental conditions are well expected and have been captured by most global models. The large spatial variability of aerosols has also been observed by numerous satellites in the past 30 years.

As stated above, the reviewer does not seem to understand all the data being described here. We hope the new manuscript is clearer.

In addition, satellite datasets are one valuable source of information, but climate scientists are interested in using multiple lines of evidence in order to assess changes in climate and radiative forcing, so here we provide another source of data. In addition, satellite datasets have no information about composition, nor are they able to see through clouds, and have a minimum detection limit that can be 50% of the signal in remote regions (0.03-0.05). We explain more

clearly in the introduction the value of independent surface concentration datasets. We clarify with the following text in the introduction (lines 154-180):
"Climate modelers are constantly looking for multiple independent lines of evidence to verify their models, and in situ surface concentration data presents a valuable source of information about aerosols often near human society.  Understanding spatial variability in aerosols, and the composition of those aerosols is key to understanding how aerosols in different regions have evolved in the past, and how they will evolve in the future. Some regions are dominated by fossil fuel derived aerosols, which may have peaked in magnitude, even as greenhouse gas concentrations continue to increase, while in other regions aerosols are driven by agriculture or by natural aerosols (Bauer et al., 2016; Turnock et al., 2020; Kok et al., 2023).  In addition, different aerosol species have different impacts on climate: for example, knowing whether aerosols are scattering or absorbing changes the sign of the interaction (Li et al, 2022).  Some aerosols also serve as better cloud or ice nuclei than others, while biogeochemical impacts are very sensitive to composition (Mahowald et al., 2011). Knowing even  the order of magnitude in regions with aerosols (e.g., contrasting 0.1 to 0.001) is important for aerosol-cloud interactions that can be non-linear especially at low aerosol levels (Carslaw et al., 2013). Having surface concentration observational dataset with large spatial coverage based on independent data can be valuable for aerosol model comparisons, especially for models with a global domain. We focus most of this paper on the spatial distribution of climatological mean, as that is easily obtained from models, and the most important variable for many climate impacts like radiative effects or aerosol-cloud interactions, except for aerosols dispersed by large infrequent events (e.g., Clark et al., 2015; Fasullo et al., 2022). Since aerosols are thought to cause between 2 and 10 million deaths per year (Landrigan et al., 2018; Lelieveld et al., 2019; Murray et al., 2020; Vohra et al., 2021), understanding and being able to model correctly the annual mean aerosol concentrations in the surface layer is vital and thus this dataset provides valuable information towards understanding aerosol contributions to mortality.  Nonetheless, there have been trends in emissions especially of anthropogenic aerosols over the last 40 years (Quass et al., 2022; Bauer et al., 2022), and we consider these as well.

For this study we focus on the following: a) identifying and compiling available $PM_{2.5}$ and $PM_{10}$ aerosol data, including aerosol composition, into a new publicly available database (AERO-MAP) for the modelling community across as much of the globe as possible; b) presenting a methodology to compare the spatial distribution of the climatological mean observations to the aerosols in an Earth system model; c) briefly present some temporal trends and comparisons available from this dataset and d) identifying the measurement and modelling gaps from this comparison.  While our model evaluation is not exhaustive, we hope that the convenience of this observational compilation enables an expanding and more thorough set of comparisons by future investigators.

        "

Therefore, I don't understand the rationale for spending tremendous effort to generate a dataset with such limited usefulness that hides interannual and seasonal variabilities and not suitable for any air quality related studies.

It is a bold assertion that climatological annual average surface concentrations are not valuable for air quality studies, and this statement is unsupported and contradicts the main air quality impacts on health study (e.g., Disease burden studies).  But we have added to the posted zenodo dataset (this was always available by request) the annual average across time, a decadal average and monthly average concentrations to the zenodo posted dataset.   All other time periods can be obtained by request to the authors (as indicated in the data availability statement), since we cannot post some data.  We rewrite this section to highlight the importance of surface concentrations for air quality (lines 168-172):

"Since aerosols are thought to cause between 2 and 10 million deaths per year (Landrigan et al., 2018; Lelieveld et al., 2019; Murray et al., 2020; Vohra et al., 2021), understanding and being able to model correctly the annual mean aerosol concentrations in the surface layer is vital and thus this dataset provides valuable information towards understanding aerosol contributions to mortality.  "

In addition, the data sources can be better described to at least mention major networks and key individual datasets, describe the similarities and differences in measurement techniques.

The main networks and key individual datasets were mentioned already in the Figure 1 caption, but we add a description of the measurement types used for this study Table 1)

Table 1: Aerosol measurement types.

| Composition | Measurement Method | Variables | | Example Networks | Example Citations |
|---|---|---|---|---|---|
| Fine and Coarse | Stacked Filter Unit (SFU) | Fine, Coarse | | U. Gent | Maenhaut et al., 2002 |
| PM2.5 and PM10 | Reference Method/Federal Equivalent Method (FRM/FEM), | PM2.5, PM10 | | IMPROVE, CASNET, EMEP | Hand et al, 2019, Putaud et al., 2004 |
| PM2.5 and PM10 | Hi Vol Sampler | | | EMEP, SINCA | Putaud et al., 2004 |
| Elemental | Particle Induced x-ray emission Spectrometry (PIXE), Instrumental nuclear activation analysis (INAA) | Al, S, Na | | U. Gent, EMEP | Maenhaut et al., 2002, |
| Elemental | Inductively Coupled Plasma-Mass Spectrometry (ICP-MS) | Al, S, Na | | EMEP, SPARTAN | Putaud et al., 2004; Phillips et al., 2017 |
| Elemental | XRF | Al, S, Na | | IMPROVE, CASNET | Hand et al, 2019 |
| Chemistry | Ion Chromatography | $SO_4^{--}$, $NO_3^-$, NHr | | IMPROVE, CASNET, EMEP | Hand et al, 2019, Putaud et al., 2004 |

| Carbonaceous | Thermal Optical Reflectance | EC, OC | | IMPROVE, CASNET | Hand et al, 2019 |
|---|---|---|---|---|---|
| | Evolved Gas Analysis Non-dispersive Infrared (EGA+NDIR) | OC, EC | | EMEP | Putaud et al., 2004 |
| | | | | | |

The authors have deferred the temporally resolved data and more information to the GHOST dataset which include a subset of the PM data collected in this work, and they "hope that the GHOST effort can expanded to include more spatial variability and be maintained into the future". This should not be the reason that the temporal variability is not provided in this AERO-MAP compilation, especially such variability might be lost if the hope for the GHOST expansion does not get realized.

As indicated in the original manuscript's data availability, all of the data is available by request, although some of it cannot be re-posted. We try to make this more clear in the new manuscript (see response to the first reviewer point).

2. Model simulation and evaluation method:
The manuscript spends most extensive volume on modeling and comparison with the complied data. A few additional aerosol sources and types have been added to the CAM6 model simulated aerosols, including agriculture dust, coarse mode BC and OC, coarse ash from industrial sources, primary biogenic particles, and ammonium nitrate. However, there are several issues that need to be clarified. It is not described how many years of the model simulation is conducted to compare with the data that are averaged from different time periods from 1986 to 2023, although it is said that the model uses the 2010 emissions.
It is also not clear how ash emissions from industrial sources are implemented (are all industrial sectors emitting ash, or just certain sectors?), if all particles are converted to the aerodynamic size and how the conversion is done, etc.
We simplify the modeling to just focus on the standard model elements plus nitrogen oxides (and thus remove the elements the reviewer questions), and clarify the meteorology for the model time period as 2013-2015.

At the end, the model evaluation stops at the mode-data comparison, and there is no "lesson learned" to move forward for using the data to constrain and improve the model.

Our goal here is to show a new observational surface concentration dataset and a method for comparing it to models (using elemental relationships), that can show the problems with models. We add in a section in the summary and conclusions indicating that the dataset should be used for other models as well, and can be used in a variety of methods to evaluate and improve models. Not all uses of the dataset can be included in this first paper.

Lines 706-711): "We also present a method that is generalizable to other models that use this dataset to evaluate both mass and composition for intercomparison projects and improvements in air quality and Earth system modelsThe novel aspect of this paper is to present this compilation in an easy to use netcdf format and some example comparisons that can be used in the future to evaluate and improve model simulations for individual models or for AEROCOM intercomparisons. The underlying data could also be used for data assimilation efforts or for estimating from the observations what the contributions are from different aerosols (e.g., similar to Prank et al., 2016)."

Besides, not to show the temporal variability in different regions have really hindered the thoroughness of model evaluation.

The paper is quite long already, but we add an example of the data available in this compilation in Section 3.5 that shows time trends in the observations as well as model comparisons to the seasonal cycle in the observations. Users of the database presented here can use this data in whatever way they want; it is impossible for us to show all the possible ways in one paper.

In addition, there are many places regarding the model that need to be clarified/corrected (see specific comments below).
We remove most of the new aerosol modeling that was confusing.

For all the model vs. observation scatter plots, it is necessary to show regional statistics instead of lumping them together to better assess the regional bias. The correlation coefficients are necessary but not sufficient; the relative bias and RMSE should also be calculated and shown.

The global relative bias, RMSE were previously provided, but we add also the regional statistics to demonstrate that such comparisons can be made in future intercomparisons as well. This means the supplemental section is huge, of course, since we have 6 regions for 2 sizes for 8 different compositions. See Tables S6-S8 and Figures S3-S17.

3. Measurement and modeling gaps identified:
In the conclusion, the identified gaps for the CESM2 (or is it CAM6?) include 1) model missing 10-60% of the particulate mass due to the lack of nitrogenous particles (ammonium and nitrate) and 2) the poorly understood agricultural dust particles. However, the comparisons are showing the opposite regarding the missing mass and lack of nitrogen aerosols. Almost all the model-data comparison figures are showing that overall, the model simulated aerosol mass and composition, including ammonium and nitrate, are higher than observed, not lower, and the actual ammonium nitrate simulated by the model must be cut in half (section 2.2) to make better agreement with the observations.

The default model does not include any nitrogen aerosols: these were added for this simulation. We modify the text to make this more clear.
Lines 670-674: "In this paper, we included nitrate aerosols, which are not included in the default CESM simulations conducted for climate, and represent about 10% of the globally averaged

surface concentration mass (Table 2; Fig. S18 and S19). When we look spatially, the default particles are the dominant particles over most of the planet (Fig. 11), but in many regions for both $PM_{2.5}$ and $PM_{10}$, the default aerosol scheme includes less than 50% of the aerosol particles (Fig. 10a and c), with substantial contributions from the nitrate particles that we add to the simulation (Fig. 10b and d). "

We also remove most of the other aerosol species because we could not statistically significantly tell if they improved the simulation. Of course, we can see the NO3 easily separately, so that is easy to indicate the importance of.

As for the "poorly understood agriculture dust", the comparisons of Al (dust proxy) are done for dust emitted from all sources and no agriculture dust is evaluated; that means there is no way to particularly point to the problems in agriculture dust.

We agree with the reviewer that it is confusing to have this constituent that we cannot not show makes a statistically significant difference and we remove it from this paper.

Given the above reasons, I recommend not to publish the manuscript in the present form. Major revision should be made to better disseminate the collected dataset and to clarify the modeling and improve the evaluation matrix.

We have simplified the model, and would like to know how better to disseminate the collected data than make it available to all users?

We hope the new text clarifies the availability of the observational data as well as clarifies the modeling methodology used here.
Specific comments:
Line 76: Change 'but' to "and".
Agreed.
Line 85-87: "Most climate models exclude 10-30% of aerosol particles…": this sentence seems baseless. How many climate models have been surveyed to reach such conclusion?
We change to:

Lines 92-93". Climate models without ammonium nitrate aerosols omit ~10% of the global average mass of aerosol particles in both $PM_{2.5}$ and $PM_{10}$ size fractions, with up to 50% of the particles not included in some regions. "
Line 112-113: "However, most of these comparisons include data only from North America and Europe": This is certainly not true. The comparisons were done with data from the globe without regional preference, although most surface PM data measurements are available from North America and Europe.
We rewrite to (lines 120-121): " However, most of the available data comes from North America and Europe (e.g., Szopa et al., 2021; Reddington et al., 2017). "

Line 161-163: The justification of focusing on climatological mean does not sound. All models should have at least monthly means that are easily to obtain, and assessing aerosol radiative effects and aerosol-cloud interactions certainly requires time-resolved fields, not climatological means.

We include the climatological, decadal, annual and climatological monthly means, and hopefully it is clear that the reviewer could obtain other datasets, so that we think this point is addressed. We also show that the climatological means are very similar to the decadal means, and that the interannual variability is not a large source of error (Section 2.3), so we hope we have addressed this point.

Line 164: "The climatological mean is obviously less important for extreme air quality events": It is really not useful for any air quality related studies which requires high spatial and temporal resolution data, not climatological means.

We do not think the literature supports these assertions. The climatological means are used in many air quality studies, especially the contribution to disease studies from Lancet. We emphasize that point here:

Lines 168-171" Since aerosols are thought to cause between 2 and 10 million deaths per year (Landrigan et al., 2018; Lelieveld et al., 2019; Murray et al., 2020; Vohra et al., 2021), understanding and being able to model correctly the annual mean aerosol concentrations in the surface layer is vital and thus this dataset provides valuable information towards understanding aerosol contributions to mortality. "

Line 176: How does advertisement at international meetings collect data? It sounds strange.

We rewrite to: "Datasets were identified by advertising at international meetings (Wiedinmyer et al., 2018), searching the literature, contacting principal investigators and accessing publicly available datasets."

Line 181-182: Does the PM10 in the compilation include PM2.5, or just PM2.5-PM10 (i.e., particle diameter larger than 2.5 micron and less than or equal to 10 micron)?

The PM10 in the compilation is all PM less than or equal to 10 micron, as this is the standard definition.

Line 184-185: The supplemental dataset 1 only shows the year of the measurements, no information on season/month/day.

The supplemental dataset shows the climatological annual mean observation, as indicated in the description. Information on the years covered is included in the dataset. In addition, due to the reviewer comments, we have added decadal 2010-2019, climatological month means, and annual means to the dataset, but more detailed data is available as per the data availability policy described in the paper.

Line 186-205: The different techniques and possible issues for each measurement datasets should be documented in the data archive.

We include a table of the types of measurements for different networks (Table 1) to address this point, and include citations for each dataset so that interested users can assess this information themselves.

Line 206: There is a 38-year time span of the data collected, when the emissions changed significantly around the globe. Therefore, the "climatological mean" is not meaningful.

Line 223: How much of the model-data difference might be attributed to that the model uses 2010 emission while the data spanning as along as 38 years?

It is a bold assertion that with the data here, the climatological mean is not meaningful (with no evidence) that is shown in Section 3.2 to not be true. Interannual variability is not one of the larger uncertainties in this dataset: the within year and spatial variability within one grid box are larger. We show that the climatological mean gives almost the same result as the decadal 2010-2019 mean, but is missing more data.

Lines 454-475:
"Trends in aerosols are an important scientific question, although for most of this paper we use the climatological annual mean. What if there were strong trends in the aerosols; would that lead to differences between our climatological means and what we expect for some decades? In order to assess this, we look at the individual annual means for each station with more than 8 years of data and see if the individual annual mean is ever outside of the 3x uncertainty calculated here. Out of the 13320 station datasets for $PM_{2.5}$ or $PM_{10}$ which have more than 8 years of data, only 175 (1.3%) have an annual average outside the uncertainty estimated here. Of those with a value outside the uncertainty, only 10 (<0.01%) have a statistically significant trend. This suggests that for the temporal interval we have chosen for the climatology, long term trends are not a significant source of differences in the spatial climatological dataset presented here. Nonetheless, we acknowledge that in regions where aerosol emissions increase and then decrease over our multi-decadal observational record (e.g. China), our test for trends will not reveal where the climatology over the full period is less representative of individual decades. We also supply in the compiled dataset a decadal mean for the time period of 2010-2019, which is made publicly available. A comparison of the climatological mean versus the decadal mean for the $PM_{2.5}$ and $PM_{10}$ concentrations show that for almost all locations, there is a small difference between the two values, and they lie on a one-to-one line (Figure 2d and 2e; Table S4). There are a few station datasets ( <5% ) which have a difference between the climatological mean and the decadal mean that is larger than 20%, and very few (<0.05%) have a difference which is larger than the uncertainties described in this section (factor of 3; Table S4). The biggest difference between the climatological and decadal average values is the number of station datasets and observations and thus spatial coverage: we lose between 20% and 100% of the station datasets, depending on the size and composition, when we use the decadal means (Table S5). This is because even though this is the most observed decade, still some datasets are outside this time period. In order to emphasize the spatial distribution of the datasets, and because the climatological values are so similar to the decadal means, we will show just the climatological values in the next few sections, although both are available (Supplemental dataset 1 and https://zenodo.org/records/10459654).

"

Lines 571-578: " How would these comparisons change if we used the decadal 2010-2019 averages instead of the climatological averages of the observations? As expected from the similarity between the observations averaged over these two time periods (Section 3.2; Table S4) the results do not substantially change (>20%) in most regions where there is a similar amount of data (Fig. S2a; Table S6). But for some regions and composition datasets, there is much less data (>25% less data), and in those cases, there can be large differences between using the decadal averages versus the climatological averages (Table S6). This suggests that using the

climatological averages for our comparisons for PM$_{2.5}$ allows us to include more data and evaluate more regions, without including much bias, since interannual variability is a small source of uncertainty compared to other uncertainties (Table S4).   "

Lines 620-625: "If we compared instead to the decadal averages rather than the climatological averages, we would obtain similar results in many cases (Fig. 2b; Table S8), but being limited to decadal averages reduces substantially the amount of observations available for comparison. The few regions which lose less than 25% of the data sets when we temporally limit our comparison have similar statistics similar as in the PM$_{2.5}$ comparisons. Again, this suggests that using the climatological averages includes more regions in the comparisons without evidence that it increases bias, because of the small amount of interannual variability in this data set (Section 3.2)."

If the reviewer is unconvinced or has specific uses for the data, we also supply decadal means, annual means, climatological monthly means (and the underlying dataset by request) so that other information can be asserted.

The data does not evolve much during the 38 years (as shown in Section 3.2), and interannual variability's uncertainty is smaller than the other uncertainties in the dataset. It is possible the modeled emission rates might change more, however that is beyond the scope of this presentation of a new compilation of surface concentrations to explore.

Line 224: CAM6 or CAM5?
The model used here is the CAM6, but the size parameters for the coarse mode were returned to those used in the CAM5.  We rewrite the sentence to be more clear.

Lines 265-268 "For this study, the coarse size mode (mode 3) was returned to the size parameters used in the previous version of the model: CAM5 (geometric standard deviation of 1.8) to better simulate coarse mode particles, and improve the dry deposition scheme and optics used in the model for simulating coarse mode particles like dust as described in Li et al. (2022). "
Line 230-231; CMIP6 emission does not include natural emissions.

Good point.

Lines 272-273: We rewrite to: "Anthropogenic emissions of sulfate, OM, and BC follow the Climate Model Intercomparison Project 6 historical data for 2010 (Gidden et al., 2019).  ."
Line 240: What crop area data was used?
We remove the agricultural dust from this paper.

Line 243: There is no separate evaluation of agricultural dust presented in the manuscript.
Yes, this is true. We could not see a difference statistically significantly, and it seemed too much for this paper, so we removed it.

General question: Does the model include secondary organic aerosols from anthropogenic, biomass burning, and biogenic VOC oxidations?

Good question.

Lines 262-264: We add: "The model includes separate primary and secondary organic species which are both emitted directly, but the primary organic and black carbon aerosols are allowed to age in the model from hydrophobic to hydroscopic, and their optical properties also change (Liu et al., 2016). "

Line 269-270: If the model already assume that sulfate and nitrate are fully neutralized as ammonium sulfate and ammonium nitrate, what is the role of simulated NH4+?

The default model includes only sulfate, which is assumed to be neutralized with ammonium. The simulated NH4+ is compared to the sulfate values, and any left over NH4+ is allowed to form ammonium nitrate. We add (line 287-288):" Ammonium nitrate is assumed to only form when there is surplus ammonium (and nitrate) after the ammonium sulfate is formed. "

Line 278-279: Such adjustment for nitrate (cut in half) does not seem to be warranted. By the same practice, one could have multiplied any number to all model simulated aerosol species for the purpose to best match the observations; this is not called model evaluation.

Good point: we remove this 'tuning' of the data.

Line 283: How does model convert sea salt to sodium and chloride? In fact, chloride is not used anywhere in the paper.

Good point: we removed the comparisons with Cl, but accidentally left in one statement with it still there. Thank you for identifying. We add at line 301-304:" We use the mean Na amounts in sea salt (31%; Schlesinger, 1997) to characterize the Na amounts and include the soluble Na measurements as well (Na$^+$) if available when Na measurements are not available. Note that Cl cannot be used to evaluate sea salts as the Cl is degassed from aerosols, primarily due to sulfate interactions (e.g.,Pio and Lopes, 1998). "

Line 294: Should be "one-to-one", not "on to one".

Thank you.

Line 301-317, the paragraph about aerodynamic size:
- since the measurements are done with a variety of techniques, how do you first harmonize the aerosol particle size with those different measurements? Clearly, some (probably most) of the technique measures aerodynamic size but others are not.
- Converging geometric diameter to aerodynamic diameter requires the consideration of both particle density and shape factor. In that regard, not only model simulated dust particles should be converted to aerodynamic size but also other aerosol species. But the aerodynamic-equivalent geometric size for these aerosols are not described in the manuscript. Are they considered? What is the equivalent sizes for these aerosols?

Good points. We remove the model adjustments for PM6.9 and some of this discussion as too much detail for this paper. We add more on aerodynamic size in measurement-model comparison uncertainty instead.

Line 353-354: "this dataset presents a huge increase in the amount of data available to the aerosol modeling community": Do you have any statistics support such statement? What is the amount of data currently available and how much increase does the data compiled in this study offer?

We clarify (line 398-401): "While this dataset presents a huge increase in the amount of data available to the aerosol modelling community (for example, an eight-fold increase compared to the datasets included in Reddington et al., 2017), still the dominant proportion of the total $PM_{2.5}$ or $PM_{10}$ data are clustered over a few industrialized land regions, and there is little composition information over most of the globe (Fig. 1)."

Line 360-361, different measurement durations: It is important to document the durations of each data to understand the difference between the model and data.
This is a good point: we add some discussion in the introduction of the data about the duration of the measurements and how this information is contained for each dataset. Since we are averaging both data and measurements over longer periods than either were originally at, it is unlikely this adds additional uncertainty. In the supplemental data we also add a variable for each composition indicating the years of measurement.

Section 3.2 in general: It is not clear what the definition of "uncertainty" is. It seems the term "variability" and "uncertainty" sometimes are used interchangeably. For example, how do you assign the uncertainty values for "within grid variability", "within year variability", and interannual variability"? (Figure 1g).
Good point: We add at line 413: "Uncertainties in the observation-model comparisons can include both uncertainties in the observations, as well as interannual variability in both the model and observations that are temporally averaged. "
Line 408-409: This sentence is unclear. Are you saying that the model overestimates SO2 compared to the satellite remote sensing SO2, so the model overestimation of PM2.5 over China can be attributed to an overestimation of emission?
Good point: we rewrite at line 497: "It seems likely that at least some of these errors are due to an overestimate in the emission databases, since satellite based remote sensing has suggested that models overestimate in $SO_2$ over China (Luo et al., 2020). "

Line 410-411: do you see from the CMIP6 emission dataset if the emissions over India and China in more recent years are lower than those in 2010 to support this statement?
Good point: We rewrite at line 475: "In addition, these discrepancies could be due to an error in the aerosol transport modelling or the time period: the observations are more recent while the assumptions for the emissions are for the year 2010 (Quass et al., 2021)."
Line 414: "Much of the data in those regions are not usually included in compilation of data": Is there any references to support that statement?
We add a reference a very solid intercomparison dataset which does not use this data as an example: "Much of the data in those regions are not usually included in routinely used previous compilations of data (e.g., Reddington et al., 2017), so the fact that previous model studies have not been able to assess emission datasets in these regions could also partially explain this discrepancy."
Line 427: What is the default version of the model and what are included?
We rewrite (line 517): "Next, we consider the composition of the $PM_{2.5}$ aerosol in the model versus the observations."
Line 429: "…but this bias (overestimation of sulfate) was seen in this model": Is there any explanation of such long history of the model high bias? I wonder why this model has not been improved since the problem has been known for more than 10 years.

Good point we add at Line 520: "Previous studies have compared $SO_4^{2-}$ aerosol observations to some model simulations and have not noted this bias (e.g., Barrie et al., 2001; Aas et al., 2019) but this bias was seen in this model and attributed to the simple chemistry included in the model (Liu et al., 2012; Yang et al., 2018). "

Line 439-440: "…(bias) must be due to biomass burning and/or industrial emissions": What diagnostics have you run to attribute the cause of the bias solely to emissions but not to chemical formation or dry/wet deposition processes?

Good point: we remove this sentence and the discussion of primary biogenics, as they are not in the study.

Line 441: Again, how is the model simulated sea salt converted to Na?
We added this in the model description part and thus do not add here.

Line 443-444 and Figure 4: If Na represents sea salt, how can it go so far inland? What is the land source of Na? Even though the industrial source of Na is not included in the model, it should be captured by the measurement over land.
Good point. We add at line 538: " Notice that we do not include industrial emissions of Na, but the concentrations far inland include some Na, suggesting land-based natural or industrial sources. "

Line 449-461 about nitrogen aerosol species: In the model description section, it is assumed that all nitrate aerosols is in the form of ammonium nitrate and all sulfate aerosol is in the form of ammonium sulfate, which contradicts the description in this paragraph that the cations (NH4+) and anions (SO4=, NO3-) are not balanced, although you do not have the thermodynamic equilibrium process to partition the amount of NH4+ to sulfate or nitrate. So, what is the point of the discussion?
We are just reminding the reader of the very simple approach used here, and the basics of nitrogen aerosol stoichiometry. We think this description is identical to the one in the methods. We try to rewrite to make more clear at line 551: "  To summarize these complicated interactions, because $SO_4^{2-}$ is a stronger acid than $NO_3^-$ in the atmosphere, the basic $NH_4^+$ is preferentially found with $SO_4^{2-}$.  Thus $NO_3^-$ particles will only form if there is sufficient $NH_4^+$ available.  "

Line 462-463: The NO3- from the model compared in Figure 4k and l is reduced in half, per description in line 278-279, right? It should be clarified.
We have removed the tuning of the modeled NO3.

Line 465: It is not clear where the "agriculture regions" are in Figure 4k and l to show the importance of NH4+.

We clarify at line 566-568: " The model and data distribution of $NH_4^+$ show the high values of $NH_4^+$ over agricultural regions especially (e.g., Vira et al., 2022), like the mid-western US or central Europe (Fig. 5m and Fig. 5n; correlation coefficient=0.52).  "

Line 473: Again, the conversion between aerodynamic diameter and geometric diameter depends on the particle composition (density, shape) and is not universal.

Line 476-477: The conversion factor for coarse sea salt can be calculated from sea salt aerosol density and shape factor.
We remove this part of the paper.

Line 478, 480, and Figure 5: The figure labels are inconsistent. Is it Fig 5a, c, d, or Fig 5a, b, c?
Good point: we fix this.
Line 485 and Figure 6a, b: How was the sulfate PM10 calculated? It was not described before.
The MAM4 model includes coarse mode sulfate. We add this to the model description, line 265:
"The coarse mode is included for sulfate, dust and sea salts."
Line 494: "well simulated" is subjective. What is the criterion for "well simulated"? Please be quantitative and objective.
We rewrite to line 605-612 "Comparisons with Al (used here as a proxy for dust) show that the spatial variability is correlated between model and observations (correlation coefficient of 0.46), but the model overpredicts the concentrations in high dust regions and underestimates in low dust region (Fig 7i and 7j; 54% of the observations are outside the uncertainty bounds in Table S7). The largest overestimates are in Asia and Africa (Fig 7i and 7j). Dust models are compared against aerosol optical depth, deposition and surface concentrations and it is currently not possible to simulate all of these different types of measurements at the same time, consistent with previous studies with this model (Li et al., 2022; Kok et al., 2014b; Albani et al., 2014a; Matsui and Mahowald, 2017; Zhao et al., 2022), and indeed across most dust models (Huneeus et al., 2011)."

General comments for section 3.3 and 3.4, mode-data comparisons: The model simulated PM2.5, PM10, and related aerosol species are compared to the compiled observations to show the model performance against the data. However, the effort seems to stop just here; there is no effort to show how the data can be used to "constrain" the model, even though it has been stated "Our goal in this study was to provide observational constraints" (line 358). What should be done next? How can the model be meaningfully constrained by observations?

This study is to show a new compilation and how it can be used, so we are not going to make model modifications. We add in text in the summary and conclusions that this data can be used for future model-data comparisons and future model developments to improve models. We add in the conclusions (line 707-713): ". The novel aspect of this paper is to present this compilation in an easy to use netcdf format and some example comparisons that can be used in the future to evaluate and improve model simulations for individual models or for AEROCOM intercomparisons. The underlying data could also be used for data assimilation efforts or for estimating from the observations what the contributions are from different aerosols (e.g., similar to Prank et al., 2016)."

Line 526: "nitrogen aerosol emissions": NH4+ and NO3- are secondary aerosols that are not directly emitted but are formed in the atmosphere from the oxidation of their precursor gases.
Good point. We rewrite (line 704): "Knowing what particles are dominant in a region is required, as fossil fuel derived aerosols will likely be reduced, while agriculturally based aerosols may well increase (Gidden et al., 2019)."
Figures:
- Figure 1g: What is the x-axis?
  Good point: relative error added to the plot.

- Figure 2a: It is very hard to see the model simulated color over North America, Europe, and Asia because they are all covered by the circles. The map should be made much larger. Currently it is illegible.
- We have added area-focused plots in the supplement. The main point of the main text is showing where there is data and where there is not data.
- Lines 479-486: "Because the high number of observations in some parts of the world would make the figures unreadable, the observations are gridded onto an approximately 2°×2° grid for comparisons with the model, although individual data points are still difficult to read (Fig. 3a). The maps illustrate where the observational comparison in the scatter plot is made, and focused maps of each major region are available in the supplement (Figure S1) as well as global and regional statistics (Table S5)."
- Figure 2b: The spatial gradient from observations are much more smaller than that from model simulations with most data fall into 10-50 micron. Increase the resolution of color contours may help reveal better the spatial gradient from the data. Also, the domain in Fig. 2b is not just East Asia; it covers Central Asia, East Asia, South Asia, and Southeast Asia. Call it Asia is more appropriate.
  Good point: We modify the title of the figure to show Central Asia. We tried changing the color bars for different subsections of the map, but think it would be confusing to modify the color bar for each region one section, so we keep the same color bar for the global and regional graphs..
- Figure 2c (and similar for other scatter plots in this paper): It is hard to distinguish the regions. I suggest plot each region separately to show the characters, and to include correlation coefficients, relative bias, and RMSE on each plot for each region.
  Our goal in this paper is to highlight how much surface data there is, so we focus on global plots in the main text. We add in a bold black symbol to make the areal averages easier to see. In order to not make the main text too big, but accommodate the reviewers request, we add in focused areas in the supplement, and add in the statistics into the supplemental tables for the region (they were already there for the global study).
  Figure 3: It is not mentioned and used in the text.
- Figure 3 (now 4) is referenced in the discussion of Figure 2 (now 3) in Section 3.3
- Figure 5: a, c, d on the panel but a, b, c, in the caption (and in the main text). Make them consistent!
- Thank you: this is corrected.
- Figure 6b title: It should be PM10, not PM2.5.
- Thank you: this is corrected.
- Figure 6, the right panel on the second row: It should be panel d, not c, and the title should be PM10, not PM2.5.
- Thank you: this is corrected.

**Citation**: https://doi.org/10.5194/egusphere-2024-1617-RC1

Reviewer #2:
Review Mahowald et al Aero-Map
General comments
The study is a very worthy attempt to gather data on PM and their composition from world wide sites. The authors are praised for the attempt and the work which went into it.

Thank you for acknowledging the huge amount of work already contained in this paper.

However, unfortunately the compilation lacks quite some detail to convince others to use it, and is a bit too vague in the discussion or errors and model-data comparison, to my opinion. Can this be improved?
The period 1986-2023 brings along considerable trends in PM concentrations, with declines of the order of 2-3% per year, amounting to reductions by probably more than 50% in three decades in many stations.
What the reviewer asserts about trends being important could be true for other datasets (although the reviewer does not indicate what dataset this is true for), but does not seem to be the case for this dataset (new text and analysis below). Notice that 50% change (which is seen very rarely in this dataset) is still within the uncertainty we estimate for the climatological average in any case (See Section 3.2).

In any case, we add to the supplemental data (and the netcdf file on zenodo) decadal average data for 2010-2019 and show a comparison of the 2010-2019 to the climatological averages in Section 3.2, showing that these are only in a few places likely to lead to errors. We also add to the zenodo netcdf file annual averages for all the years available, so that the reviewer (and other dataset users) are free to choose whatever time period they would like.

We add into the paper:
 Lines 454-475:
"Trends in aerosols are an important scientific question, although for most of this paper we use the climatological annual mean. What if there were strong trends in the aerosols; would that lead to differences between our climatological means and what we expect for some decades? In order to assess this, we look at the individual annual means for each station with more than 8 years of data and see if the individual annual mean is ever outside of the 3x uncertainty calculated here. Out of the 13320 station datasets for $PM_{2.5}$ or $PM_{10}$ which have more than 8 years of data, only 175 (1.3%) have an annual average outside the uncertainty estimated here. Of those with a value outside the uncertainty, only 10 (<0.01%) have a statistically significant trend. This suggests that for the temporal interval we have chosen for the climatology, long term trends are not a significant source of differences in the spatial climatological dataset presented here. Nonetheless, we acknowledge that in regions where aerosol emissions increase and then decrease over our multi-decadal observational record (e.g. China), our test for trends will not reveal where the climatology over the full period is less representative of individual decades. We also supply in the compiled dataset a decadal mean for the time period of 2010-2019, which is made publicly available. A comparison of the climatological mean versus the decadal mean for the $PM_{2.5}$ and $PM_{10}$ concentrations show that for almost all locations, there is a small difference between the two values, and they lie on a one-to-one line (Figure 2d and 2e; Table S4). There are a few station datasets ( <5% ) which have a difference between the climatological mean and the decadal mean that is larger than 20%, and very few (<0.05%) have a difference which is larger than the uncertainties described in this section (factor of 3; Table S4). The biggest difference between the climatological and decadal average values is the number of station datasets and observations and thus spatial coverage: we lose between 20% and 100% of the station datasets, depending on the size and composition, when we use the decadal means (Table S5). This is

because even though this is the most observed decade, still some datasets are outside this time period. In order to emphasize the spatial distribution of the datasets, and because the climatological values are so similar to the decadal means, we will show just the climatological values in the next few sections, although both are available (Supplemental dataset 1 and https://zenodo.org/records/10459654).”

Lines 571-578: “How would these comparisons change if we used the decadal 2010-2019 averages instead of the climatological averages of the observations?  As expected from the similarity between the observations averaged over these two time periods (Section 3.2; Table S4) the results do not substantially change (>20%) in most regions where there is a similar amount of data (Fig. S2a; Table S6).  But for some regions and composition datasets, there is much less data (>25% less data), and in those cases, there can be large differences between using the decadal averages versus the climatological averages (Table S6).  This suggests that using the climatological averages for our comparisons for $PM_{2.5}$ allows us to include more data and evaluate more regions, without including much bias, since interannual variability is a small source of uncertainty compared to other uncertainties (Table S4).   “

Lines 620-625: “If we compared instead to the decadal averages rather than the climatological averages, we would obtain similar results in many cases (Fig. 2b; Table S8), but being limited to decadal averages reduces substantially the amount of observations available for comparison. The few regions which lose less than 25% of the data sets when we temporally limit our comparison have similar statistics similar as in the $PM_{2.5}$ comparisons. Again, this suggests that using the climatological averages includes more regions in the comparisons without evidence that it increases bias, because of the small amount of interannual variability in this data set (Section 3.2).”

If the reviewer is unconvinced or has specific uses for the data, we also supply decadal means, annual means, climatological monthly means (and the underlying dataset by request) so that other information from this dataset can be used.

That would make any average strongly dependent of the period averaged. A better documentation of the sampling period is needed for usefulness. How large is the temporal coverage of the data in the period used for averaging? How large is the temporal coverage for the chemical composition data put aside the PM data? (if that
s not the same, I would recommend to produce different tables or temporal coverage and period for each component to be specified).
These are good points. We add the Figure 2a and 2b to show the temporal coverage in each of the regions for the PM2.5 and PM10 data.  We add in the annual average data for each year, as well as a decadal average for 2010-2019 (chosen to have the most data).

We also compare the results using a climatology versus using the decadal average (as indicated in the answer to the point above).

There is a vague description of taking in more data, if the region is scarcely sampled in the paper.  Which data are used, although they do not fulfill the general coverage requirements set out in the paper?
We include TSP data in some places that have no other data (e.g., Africa): we clarify the text. The specific station datasets which are TSP data are included in the supplemental data and the netcdf files.
Line 193-194: "In some poorly measured regions, we include total suspended particles (TSP) datasets (information on the size fraction measured is in the Supplemental dataset)."

Even though the spatial variability is larger than the trend, the comparison in a region is dominated by the emissions in a specific period under consideration. The model data comparison in this paper also demonstrates that regional bias is to be understood and not just the global spatial sd.

We are not sure we understand the comment here, but include annual averages for each year, decadal averages over 2010-2019, and monthly means.  We also add in more details about the regional biases and plots in the supplement, although it is beyond the scope of this study to go into substantial detail about each region. See our response to a similar point above.
A little more cares needs to go into preparing the main output, the yearly averages in csv format: Why do coarse and fine data appear? I thought all is converted to PM10 and PM25 ? It left me wondering if data are doubled. I found for instance PM10, PM25 and coarse and fine for Birkenes, Norway, for slightly different, but overlapping periods.  Also station K_puszta. Hard to check throughout for users/me.
These are good questions: we add in more details in the manuscript to explain why there are repeats of the station name. lines 228-245:

"Our goal is to create easy-to-use datasets for model-data comparisons. Included in this dataset are several files with different levels of description and analysis.  One file provides traceability information, including a detailed citation, type and number of measurements included, as well as time period, climatological and decadal (2010-2019) means and standard deviations for each time period (Supplemental dataset 1).  For each station dataset included in the database, there will be one line in this file.  This means that for some stations (for example K-puszta), there are multiple lines in the supplemental file indicating the two different time periods where measurements were made as well as the two sizes that are measured during each time period. For each station dataset, there are latitude, longitudes, annual mean values, number of observations, year extent of the observations, standard deviations, etc, as well as the citation and where to obtain the data. There are also several netcdf files available at https://zenodo.org/records/11391232 for this dataset. The most useful is likely to be the Allobservation.AEROMAP.nc file, which contains the same quantitative data for each station dataset as the supplement, except that the data is processed to be only $PM_{2.5}$ and $PM_{10}$ (with some TSP data in places with little data, as discussed above). That means $PM_{2.5}$ and coarse aerosol mass are added together if the station datasets are collocated to create a PM10 dataset (e.g., see Table 1). In addition, this file contains climatological monthly means, and annual means for each year for each station dataset, so that temporal information is also easily available. Another file includes the climatological mean observations averaged up to a 2°×2° grid that is

used for plotting the figures shown in the paper (Allobservation.AEROMAP.2x2.nc). As indicated in the data availability, only the time-means are available and the underlying data for some datasets cannot be openly published, but please contact the authors (identified by the citation) if other time periods are desired"

What is the PM measurement method used, can the PM methods be categorised? That categorisation could contain info on a certain likely bias.

Good point.  We add in more details on the PM measurement method in the methods section and add Table 1.

Table 1: Aerosol measurement types.

| Composition | Measurement Method | Variables | | Example Networks | Example Citations |
|---|---|---|---|---|---|
| Fine and Coarse | Stacked Filter Unit (SFU) | Fine, Coarse | | U. Gent | Maenhaut et al., 2002 |
| PM2.5 and PM10 | Reference Method/Federal Equivalent Method (FRM/FEM), | PM2.5, PM10 | | IMPROVE, CASNET, EMEP | Hand et al, 2019, Putaud et al., 2004 |
| PM2.5 and PM10 | Hi Vol Sampler | | | EMEP, SINCA | Putaud et al., 2004 |
| Elemental | Particle Induced x-ray emission Spectrometry (PIXE), Instrumental nuclear activation analysis (INAA) | Al, S, Na | | U. Gent, EMEP | Maenhaut et al., 2002, |
| Elemental | Inductively Coupled Plasma-Mass Spectrometry (ICP-MS) | Al, S, Na | | EMEP, SPARTAN | Putaud et al., 2004; Phillips et al., 2017 |
| Elemental | XRF | Al, S, Na | | IMPROVE, CASNET | Hand et al, 2019 |
| Chemistry | Ion Chromatography | SO4--, NO3-, NHr | | IMPROVE, CASNET, EMEP | Hand et al, 2019, Putaud et al., 2004 |
| Carbonaceous | Thermal Optical Reflectance | EC, OC | | IMPROVE, CASNET | Hand et al, 2019 |
| | Evolved Gas Analysis Non-dispersive Infrared (EGA+NDIR) | OC, EC | | EMEP | Putaud et al., 2004 |
| | | | | | |

What shall the user of the dataset understand from when min year and max year has values of 0, -999?

If the min year and max are 0 or -999 there is an error in the spread sheet: thank you for point this out. We add minimum and maximum year for all variables to be more complete.

How is the standard deviation to be understood (is that from yearly? means over the period of observation)?
We clarify this at line 229. "One file provides traceability information, including a detailed citation, type and number of measurements included, as well as time period, climatological and decadal (2010-2019) means and standard deviations for each time period (Supplemental dataset 1). "

Why is EC and BC included, Why OC and OM? Why S and SO4? What is Nap? Why Al, dust and ash? Would it be easier to have a metadata column signalling methods?

We include all available measurement data, and in the using the variable name as submitted to make it more traceable. We include a table in the supplement which indicates how to map from the model variables to the measured variables, and clarify these points (Existing Table S2). At Line 228: "Our goal is to create easy-to-use datasets for model-data comparisons. Included in this dataset are several files with different levels of description and analysis. One file provides traceability information, including a detailed citation, type and number of measurements included, as well as time period, climatological and decadal (2010-2019) means and standard deviations for each time period (Supplemental dataset 1). For each station dataset included in the database, there will be one line in this file. This means that for some stations (for example K-puszta), there are multiple lines in the supplemental file indicating the two different time periods where measurements were made as well as the two sizes that are measured during each time period. For each station dataset, there are latitude, longitudes, annual mean values, number of observations, year extent of the observations, standard deviations, etc, as well as the citation and where to obtain the data. There are also several netcdf files available at https://zenodo.org/records/11391232 for this dataset. The most useful is likely to be the Allobservation.AEROMAP.nc file, which contains the same quantitative data for each station dataset as the supplement, except that the data is processed to be only $PM_{2.5}$ and $PM_{10}$ (with some TSP data in places with little data, as discussed above). That means $PM_{2.5}$ and coarse aerosol mass are added together if the station datasets are collocated to create a PM10 dataset (e.g., see Table 1). In addition, this file contains climatological monthly means, and annual means for each year for each station dataset, so that temporal information is also easily available. Another file includes the climatological mean observations averaged up to a 2°×2° grid that is used for plotting the figures shown in the paper (Allobservation.AEROMAP.2x2.nc). As indicated in the data availability, only the time-means are available and the underlying data for some datasets cannot be openly published, but please contact the authors (identified by the citation) if other time periods are desired"

The paper was sent to ESSD before submission to ACP. I have looked at the many comments from two reviewers there, and the short response to these comments by the authors. The authors stated that all would be considered before re-submission to ACP. Although some changes are introduced to the paper, it is very unclear to me, what has been addressed, and my impression is that not all comments have been taken up. I would like to ask the authors to address all relevant

comments from the ESSD review as well here. Its a waste of time for reviewers and readers, if those valuable comments are lost in the transfer to another EGU paper. I understand that the comparison to the model was not fit for ESSD, but that was only part of what the reviewers asked/commented on.

We add here in a separate document, the comments from the reviewer sand how the comments were addressed briefly.

I finally wonder why the annual mean is used and not a monthly mean climatology. Its not so much more data and it would make the data much more useful.
Good idea! We add in the monthly mean climatology. We show just a few comparisons with the monthly mean, just because of lack of space in this paper.

And how much of the PM10+PM25 mass is reconstructable from the composition? I cant find an evaluation of that. Would be useful to underpin that there is some missing component. I actually do not think its nitrate, or spores, or ash… what about water?
Unfortunately further analysis of each of the datasets is beyond the scope of this study, although it has been done for some datasets already separately: we add this as a good topic for further study. Studies have previously shown that for some datasets water is important (as discussed in the uncertainty analysis, since some aerosol measures are dry and some include water).
Line 296: "Modelled values of PM content, which assume dry particles, are used here, while gravimetric measurements in some networks are equilibrated at 50% relative humidity, thus 5-25% of the mass of measured PM can be water (Prank et al., 2016; Burgos et al., 2020)."

Specific comments
The 2x2 aggregated file contains per lat lon box the number of observations and the number of stations. What is number of observations here? I cant find a clear definition of how this is counted.
Good point: We add more information into a readme file, as well as the description of the datasets in section 2.1 (repeated above).

Lines337: "In this dataset, the number of station datasets included in the average is included (stations) and the number of observations add up across all the station datasets included."

l86 Most climate models exclude 10-30% of the aerosol particles in both PM2.5 and PM10 size fractions
=> where does this number come from?
This comes from Table 2 and Figure 10. We rephrase (line 92) ". Climate models without ammonium nitrate aerosols omit ~10% of the global average mass of aerosol particles in both $PM_{2.5}$ and $PM_{10}$ size fractions, with up to 50% of the particles not included in some regions."
l148 : "here we focus on characterizing in observations and models the spatial variability of the surface concentrations "
" => grammar wrong?
We rewrite the sentence:
l161 We focus on the spatial distribution of climatological mean, as that is easily obtained from models, and the most important variable for many climate impacts

=> is it really true, that the mean spatial distribution is most important?
We rewrite the sentence.
Line 155-161: "Understanding spatial variability in aerosols, and the composition of those aerosols is key to understanding how aerosols in different regions have evolved in the past, and how they will evolve in the future. Some regions are dominated by fossil fuel derived aerosols, which may have peaked in magnitude, even as greenhouse gas concentrations continue to increase, while in other regions aerosols are driven by agriculture or by natural aerosols (Bauer et al., 2016; Turnock et al., 2020; Kok et al., 2023). In addition, different aerosol species have different impacts on climate: for example, knowing whether aerosols are scattering or absorbing changes the sign of the interaction (Li et al, 2022). "

l166 Quass et al., 2022 => Quaas et al.
We correct.
l181 "Some measurement sites measure PM2.5 and coarse (PM2.5 to PM10) aerosols "
=> Do they really measure that, or do they only provide that coarse aerosol info as difference of two measurements ? See also my comment above to coarse and fine PM data.
Yes, some measure PM2.5 and coarse and some PM2.5 and PM10. The new Table 1 should clarify this.
l183 we included less complete datasets at sites in regions with limited data. =>
What does that exactly mean? Which data are included thus? Please mark in the table.

We clarify this statement. Line 190-194: "Some measurement sites have only a few observations of composition or mass, while others have multiple years: we included less complete datasets at sites in regions with limited data (e.g., field data: these are identified as station datasets with less than one year of data in supplemental datasets). In some poorly measured regions, we include total suspended particles (TSP) datasets (information on the size fraction measured is in the Supplemental dataset). The time period for different datasets is included in the supplemental dataset 1. "
l184 The time period for different datasets is included in the supplemental dataset 1
=> I only find the time period for PM. Are the composition measurements really done for the same period as for PM?
Good point: We include information for each constituent to better clarify the dataset.
l196-197: we include measurements of total suspended particulates (TSP) with the PM10, because of the lack of size-resolved data.
=> What does that exactly mean? Which data are included thus? Please mark in the table.
This information is already marked in the supplemental table and the netcdf files.
l201: wrt to the definition 1.8 * OC = OM => Why is this mentioned? Why are both OC and OM data in the table?
We clarify this information: we use both OC and OM measurements to compare to the model output. Usually only one or the other is measured, and we use these commonly used conversions to go from one to the other. Lines 229-239.
"One file provides traceability information, including a detailed citation, type and number of measurements included, as well as time period, climatological and decadal (2010-2019) means and standard deviations for each time period (Supplemental dataset 1). For each station dataset included in the database, there will be one line in this file. This means that for some stations (for example K-puszta), there are multiple lines in the supplemental file indicating the two different

time periods where measurements were made as well as the two sizes that are measured during each time period.  For each station dataset, there are latitude, longitudes, annual mean values, number of observations, year extent of the observations, standard deviations, etc, as well as the citation and where to obtain the data. There are also several netcdf files available at https://zenodo.org/records/11391232 for this dataset. The most useful is likely to be the Allobservation.AEROMAP.nc file, which contains the same quantitative data for each station dataset as the supplement, except that the data is processed to be only PM$_{2.5}$ and PM$_{10}$ (with some TSP data in places with little data, as discussed above)."

repeated in l300, can be omitted in one or the other place, anyway I dont understand exactly why this is mentioned.

l202: why include EC and BC? Are they treated differently when comparing to the model?

We include all available measurements, and thus include both EC and BC when measured. We refer to these as the original paper refers to them to make it more traceable back to the original paper.

l216+l261 : Not sure why E3SM is mentioned, can be omitted.

Because of the funding source and source of the aerosol model.

l278 the calculated nitrate aerosol amounts are multiplied by 0.5 to best match the available observations.

=> I dont see why this should be done here, why not removing all bias with a factor in preprocessing?

We agree: we have removed this.

l299 Table S2: I do not see the value of this table, can be omitted and the info is already, or should be just in the text.

We think summarizing this information in the Table is important, as it is confusing elsewise (see above the reviewer's confusion).

l294: on- to one ….

Corrected.

l303: electrical mobility analysers: when are these used for PM measurements? omit.

Corrected.

l309 and more laces: The use of PM6.9 can not be suggested like this. It really only makes sense for dusty episodes, stations. In the paper it sounds like it should be used in general.

We agree and remove the PM6.9 discussion

l318/319: For ease of viewing, ….the comparisons should also be shown for different regions in separate figures.

Because of space constraints and the sheer amount of information in this chapter we will continue to focus on the global pictures, but we add in the regions in the supplement, and the statistics in the supplemental text.  Supplemental Figures 2, 4-18.

l324: Notice that we include both urban regions and rural or remote sites

=> well, urban sites will create a negative bias for the global model. So it would be needed to identify them. Or rather state, that you do not know how exclude the urban sites.

The model includes urban areas and urban emissions so it is likely to bias the observations low if urban sites are not included.  In addition we do not know how to exclude urban sites, so we clarify this.  Line 339-344:

"Notice that we include both urban regions and rural or remote sites into the same dataset.  Some of the original meta data did not include the resolution of the location to better than 0.1 degrees,

so that the coordinates of the locations here provided with the gridded data should not be used for finer resolution studies.  Because of the importance and size of megacities, which cross multiple grid boxes, as well as the difficulty in separating urban vs. rural sites, we include urban and rural air quality data in the same dataset, and previous studies show the expected differences between urban and rural concentrations and trends (e.g., Hand et al., 2019). "

l333: with in => within ?
Thank you. Corrected.
l338 comparisonl => comparison
Thank you. Corrected.

l338 dry vs. moist aerosol mass different inlet geometries => clarify
and in general, are the methods known? Why not specify the most important measurement categories in the table?
Good idea. We add more discussion of the measurement types and include Table 1.

Table 1: Aerosol measurement types.

| Composition | Measurement Method | Variables | | Example Networks | Example Citations |
|---|---|---|---|---|---|
| Fine and Coarse | Stacked Filter Unit (SFU) | Fine, Coarse | | U. Gent | Maenhaut et al., 2002 |
| PM2.5 and PM10 | Reference Method/Federal Equivalent Method (FRM/FEM), | PM2.5, PM10 | | IMPROVE, CASNET, EMEP | Hand et al, 2019, Putaud et al., 2004 |
| PM2.5 and PM10 | Hi Vol Sampler | | | EMEP, SINCA | Putaud et al., 2004 |
| Elemental | Particle Induced x-ray emission Spectrometry (PIXE), Instrumental nuclear activation analysis (INAA) | Al, S, Na | | U. Gent, EMEP | Maenhaut et al., 2002, |
| Elemental | Inductively Coupled Plasma-Mass Spectrometry (ICP-MS) | Al, S, Na | | EMEP, SPARTAN | Putaud et al., 2004; Phillips et al., 2017 |
| Elemental | XRF | Al, S, Na | | IMPROVE, CASNET | Hand et al, 2019 |
| Chemistry | Ion Chromatography | SO4--, NO3-, NHr | | IMPROVE, CASNET, EMEP | Hand et al, 2019, Putaud et al., 2004 |
| Carbonaceous | Thermal Optical Reflectance | EC, OC | | IMPROVE, CASNET | Hand et al, 2019 |
| | Evolved Gas Analysis Non-dispersive Infrared (EGA+NDIR) | OC, EC | | EMEP | Putaud et al., 2004 |

| | | | | | |
|---|---|---|---|---|---|
| | | | | | |

l350 and other places: "total mass" (PM) =>
What to you mean TSP? PM10? PM25 or any of them?
PM just means the bulk mass in whatever size category we are discussing.  We check to make sure that we are always specifying which size in the text.
l359 prioritizing long term stations with composition data, but in regions with few measurements, we include only PM data, or data collected during field campaigns, which may last only a month or two.
=> what does that mean? Its a  vague statement. Can you add a simple categorizing column in the table, which explains why you included a station?
The information on which size is already in the tables and files, and indicates the TSP, so easy to exclude if the user prefers.  We add the following text at line 187:"Some measurement sites have only a few observations of composition or mass, while others have multiple years: we included less complete datasets at sites in regions with limited data (e.g., field data: these are identified as station datasets with less than one year of data in supplemental datasets). In some poorly measured regions, we include total suspended particles (TSP) datasets (information on the size fraction measured is in the Supplemental dataset).  The time period for different datasets is included in the supplemental dataset 1. "
l367 Thus, the dataset described here cannot do a good job of constraining aerosol concentrations that are due to episodic emission events like wildfires or dust in regions without long term datasets. =>
Probably true, but what does the user take from this statement?
We remove this sentence.
l372 "variability contribution to the uncertainties"
=> Is variability an uncertainty? Not per se, I would say. Be more precise.
Good point, we rewrite at line 413:
" Uncertainties in the observation-model comparisons can include both uncertainties in the observations, as well as interannual variability in both the model and observations that are temporally averaged.  "
l373: for within year, with grid
=> for within year, within grid  ?
Corrected.
l379 ff. What about decadal trends affecting the spatial variability, if the stations are sampled in different periods?
Trends do not appear to be important for the spatial comparison, as outlined above.  We discuss the importance of trends more in Section 3.2 and what trends are detected in Section 3.5.  We add into the datasets both 2010-2019 decadal average as well as individual years so that scientists can focus on whatever time period is most appropriate for their problem.

l385: trends can be neglected? =>
I really dont think so.
We discuss the importance of trends more in Section 3.1 and what trends are detected in Section 3.5.
Lines 454-475:

"Trends in aerosols are an important scientific question, although for most of this paper we use the climatological annual mean. What if there were strong trends in the aerosols; would that lead to differences between our climatological means and what we expect for some decades? In order to assess this, we look at the individual annual means for each station with more than 8 years of data and see if the individual annual mean is ever outside of the 3x uncertainty calculated here. Out of the 13320 station datasets for $PM_{2.5}$ or $PM_{10}$ which have more than 8 years of data, only 175 (1.3%) have an annual average outside the uncertainty estimated here. Of those with a value outside the uncertainty, only 10 (<0.01%) have a statistically significant trend. This suggests that for the temporal interval we have chosen for the climatology, long term trends are not a significant source of differences in the spatial climatological dataset presented here. Nonetheless, we acknowledge that in regions where aerosol emissions increase and then decrease over our multi-decadal observational record (e.g. China), our test for trends will not reveal where the climatology over the full period is less representative of individual decades. We also supply in the compiled dataset a decadal mean for the time period of 2010-2019, which is made publicly available. A comparison of the climatological mean versus the decadal mean for the $PM_{2.5}$ and $PM_{10}$ concentrations show that for almost all locations, there is a small difference between the two values, and they lie on a one-to-one line (Figure 2d and 2e; Table S4). There are a few station datasets ( <5% ) which have a difference between the climatological mean and the decadal mean that is larger than 20%, and very few (<0.05%) have a difference which is larger than the uncertainties described in this section (factor of 3; Table S4). The biggest difference between the climatological and decadal average values is the number of station datasets and observations and thus spatial coverage: we lose between 20% and 100% of the station datasets, depending on the size and composition, when we use the decadal means (Table S5). This is because even though this is the most observed decade, still some datasets are outside this time period. In order to emphasize the spatial distribution of the datasets, and because the climatological values are so similar to the decadal means, we will show just the climatological values in the next few sections, although both are available (Supplemental dataset 1 and https://zenodo.org/records/10459654).

"

Lines 571-578: "How would these comparisons change if we used the decadal 2010-2019 averages instead of the climatological averages of the observations? As expected from the similarity between the observations averaged over these two time periods (Section 3.2; Table S4) the results do not substantially change (>20%) in most regions where there is a similar amount of data (Fig. S2a; Table S6). But for some regions and composition datasets, there is much less data (>25% less data), and in those cases, there can be large differences between using the decadal averages versus the climatological averages (Table S6). This suggests that using the climatological averages for our comparisons for $PM_{2.5}$ allows us to include more data and evaluate more regions, without including much bias, since interannual variability is a small source of uncertainty compared to other uncertainties (Table S4).

. "

Lines 620-625: "If we compared instead to the decadal averages rather than the climatological averages, we would obtain similar results in many cases (Fig. 2b; Table S8), but being limited to decadal averages reduces substantially the amount of observations available for comparison. The few regions which lose less than 25% of the data sets when we temporally limit our comparison have similar statistics similar as in the PM$_{2.5}$ comparisons. Again, this suggests that using the climatological averages includes more regions in the comparisons without evidence that it increases bias, because of the small amount of interannual variability in this data set (Section 3.2).
"

If the reviewer is unconvinced or has specific uses for the data, we also supply decadal means, annual means, climatological monthly means (and the underlying dataset by request) so that other information can be asserted.

l394 there is much more variability across different grid boxes (4-5 orders of magnitude) than in time (up to 50%)
=> "in time" - what does that mean?
We clarify in line 439-434. "Notice that if we use the same metric (normalized standard deviation) to evaluate the variability across the climatological concentrations measured in the observations at different locations (Figure 3a) or across the grid averages in the model we obtain 1.0 and 2.2, respectively, much larger than the uncertainties (0.6): there is much more variability across different grid boxes (4-5 orders of magnitude: see Figure 2d) than across different years (up to 50% normalized standard deviation; Figure 2f).."
l394 As expected, the model contains more spatial variability than the observations,
=> Why expected ?? after colocation with the measurements? or without colocation? I think the comparison should be done after colocation.
We clarify at lines 443-446: " As expected, the model contains more spatial variability than the observations, as the model reports concentrations in very high (North Africa) and very low (Antarctica) aerosol regions where we have no data, although where we have data, the model simulates a similar range (Figure 3a). "
=> By the way, where do I see that the model has more spatial variability?
We try to be more clear: to illustrate the spatial plots have to have log-log plots to show the variability, but we don't need this for the trends (small changes—about 1%/year, not orders of magnitude per year). As above, lines 493-434: " Notice that if we use the same metric (normalized standard deviation) to evaluate the variability across the climatological concentrations measured in the observations at different locations (Figure 3a) or across the grid averages in the model we obtain 1.0 and 2.2, respectively, much larger than the uncertainties (0.6): there is much more variability across different grid boxes (4-5 orders of magnitude: see Figure 2d) than across different years (up to 50% normalized standard deviation; Figure 2f)."
l411 It would be useful to quantify which fraction of the model data points is within factor 3 of the observations.
Good idea: we add to the table in the supplement and to the discussion.
l413 The dot plots are rather un-insightful. It would be more interesting to show a map of bias.

Unfortunately, there is not enough data across the globe to show the bias: it will look really splotchy except in a few regions with a lot of data. We think that adding the regional plots in the supplement will allow details to be seen better, as well as the extra tables, and the 'average' comparisons in the main text for each region. Our goal in this study is to present the data and show how to do the model/data comparisons (convert sea salts to Na, for example), and it would be too much information to add in a detailed analysis of every region for every aerosol type.

l425 Why is the correlation coefficient going done for the gridded data? Is that an effect of the smaller weight given to well observed regions?

Yes, we add in this statement. We also switch to rank correlations, to make it so we are less biased by a few points. Line 470 " The scatterplots show the comparisons of the model to the observations using the gridded data (Fig. 3c) and all original data (Fig. 3d), and the correlation coefficients are similar (0.60 vs. 0.67 in Fig 3c and Fig 3d, respectively). It is interesting that the correlation using the ungridded data (Fig 3d) is slightly higher, perhaps because the model does better in regions with more data, although this is not a statistically significant result. The averages over different regions show that on average, the model is simulating the regions within the uncertainty (bold black symbols in Fig 3d; Table S5). "

In general: How were the uncertainties of the component concentrations established?

Good point: We calculate the uncertainties similarly to the bulk concentrations and they are similar: we add in these details. Line 446-449: For composition measurements, there is larger uncertainty in some individual species (e.g., BC and Al) than for PM. However there are many fewer composition observations (Table S3). Since the statistics of the uncertainty calculations are likely more robust with the bulk PM measurements, as there are an order of magnitude more data for the bulk PM data, we use the uncertainty estimate derived for PM for all of the composition data in this paper.

l540 Unfortunately, there are still very limited data characterizing both the surface concentration, size and composition of aerosol particles
=> How do you get to the statement, that there are very limited data available? When will we have enough data?

We rewrite to be more clear. Only a concerted effort will get us more surface concentration data, unfortunately.

Line 696: "Unfortunately, there are still very limited data characterizing both the surface concentration, size and composition of aerosol particles (Fig. 10), and the locations where we lack data have also been identified as lacking sufficient remote sensing data as well (Millet et al., 2024). "

l549 We also present a method that is generalizable to other models …
l552 This study has highlighted the value of surface concentration data
=> not sure you present a "method" and that you "highlight the value". What did you learn using the data and the method, that you did not know before about CESM?

We rewrite line 706: " We also present a method that is generalizable to other models that use this dataset to evaluate both mass and composition for intercomparison projects and improvements in air quality and Earth system models. "

We now know how the model compares to available observations, and could use this information to improve the model…

Line 707-711 ".  The novel aspect of this paper is to present this compilation in an easy to use netcdf format and some example comparisons that can be used in the future to evaluate and improve model simulations for individual models or for AEROCOM intercomparisons.  The underlying data could also be used for data assimilation efforts or for estimating from the observations what the contributions are from different aerosols (e.g., similar to Prank et al., 2016)."

line 712 "This study has highlighted the value of surface concentration data by showing that it can identify where models do well or poorly not just for total mass, but also for different compositions and size, complimenting other data sources, such as remote sensing."

l560 This study also highlights the importance of including all aerosol components into the models,

=> not sure about this one either. Where does it highlight the importance of including all components?

l561 in many places there is between 10-60% of the particulate mass missing, largely due to lack of the nitrogenous particles

=>  Where do you show that you miss 10-60% of the mass? The abstract mentioned 10-30%. Anyway, where is the statistics presented in the paper? And why is it largely due to the lack of nitrogeneous particles? What about other unknowns: SOA, water?

In this paper we have evaluated the nitrogenous particles and shown they are missing, but we have not seen missing SOA or water in this study, although they may exist in other datasets. We rewrite line 695: "This study also highlights the importance of including all aerosol components into the models, and shows that in the CESM2 approximately 10% is missing. In many places, there is 50% of the particulate mass missing, due to lack of the nitrate particles (Fig. 10; Paulot et al., 2016; Adams et al., 1999; Thornhill et al., 2020).  "

**Citation**: https://doi.org/10.5194/egusphere-2024-1617-RC2

The ACPD reviewers want responses to the reviews from the ESSD submission.  We included the revisions in the ACPD submission.  We discuss here briefly (black) how we accommodated the reviewers comments (in red).

Response to reviewers from ESSD:
Reviewer #1
"AERO-MAP: A data compilation and modelling approach to understand the fine and coarse mode aerosol composition" aims to describe a new compilation of global aerosol observations including PM2.5, PM10 and PM components (BC, OC, SO42-, Al, NO3-, NH4+, Na, and Cl).  The measurement data apparently consists of over 20 million observations taken over 15,000 stations spanning 1986-2023.  Such a dataset would be of high value to the climate and air quality modeling communities.  However the data that is under review has been temporally averaged in such a way that severely limits its use for climate or air quality model evaluation or characterization of global aerosols.  I discuss this further in my Specific Comments.
We add Section 3.2 a discussion of uncertainties to show that climatological means across spatial distributions are important and that the interannual variability is a small source of    variability.
In addition, the paper is lacking in details related to the "planning, instrumentation, and execution of collection of data" which should be included for an ESSD data description paper.  Despite the large volume of observations being presented, the observation data section consists of 4 paragraphs of the entire manuscript.  The authors provide references for the observations in one of their data files but there should be a more in depth discussion on how instrument or sampling differences across the different data sources and across years of the study period (which spans decades) could affect interpretation of the observed values.  It would also be beneficial to more explicitly state how this data compilation differs from other data sources.  The datasets themselves  have several technical issues which I discuss below.
We add in Section 2.1 more about how the data is collected and the dataset information is included in the supplemental dataset.   We add the error estimates in Section 3.2 to show the differences in measurement period and interannual variability.
The large portion of the paper is spent describing a global modeling study for 2013-2015 including detailed model specifications and an evaluation that compares the PM observed data to the model results. The authors state this component of the paper is included as "a methodology for comparing the datasets to model output, and show the implications of these results using one model."   However the observed data they use is not suitable for evaluating the global model output due to the incommensurability between the temporal averaging of the measurements and the 3 years of model simulations.
We add in the uncertainty estimates in Section 3.1. As we show in Section 3.1, the idea that the time period matters so much for the observational comparison is an unsubstantiated assertion that may be true for some other dataset (not cited) but is not true for this dataset.
 Beyond that concern, the lengthy diagnostic evaluation section describes model biases and provides possible explanations for the differences between the model and observed data.  This type of model application appears to deviate from the ESSD aim and scope for

data articles and may be better suited for a technical science article.   Specifically the results section is counter to the ESSD policy: "Any interpretation of data is outside the scope of regular articles." and "Any comparison to other methods is beyond the scope of regular articles."

Absolutely true and helpful comment and we moved the article to another journal.

The model code and output is included as part of the data package but there is not a reason given why the model output itself would be of high value to the modeling community.

It is required that the model itself be provided, so we provided it. We also doubt the value.

 I recommend the new compilation of observed data be shared with higher temporal resolution, ideally the native temporal resolution of the measurements if possible, e.g., hourly or daily averages.

We make the data available at all temporal resolutions, as stated in the data availability policy (although we can't post it).

 I also recommend that the modeling component of the paper be removed and submitted to a different journal.

We think showing how the elemental data can be used to compare to the model is helpful, so we move the whole paper to another journal.

The authors state that they plan to provide more temporally resolved observed data in a future study but this is counter to the ESSD policy to "enable the reviewer and the reader to review and use the data, respectively, with the least amount of effort. To this end, all necessary information should be presented through the article text and references in a concise manner **and each article should publish as much data as possible.  The aim is to minimize the overall workload of reviewers (e.g. by reviewing one instead of many articles)**".

We make all the data available as indicated in the data availability section of the paper (although we cannot post it all, due to restrictions.)

**Specific Comments**
My largest concern is that the aerosol measurement data being shared has been averaged across all years of available data which spans multiple decades, 1986-2023.  The only temporal information we are provided are variables called "PM_year_min" and "PM_year_max" and the number of observations taken between those two years.  I mapped what this looked like for the global measurements and found that measurements taken in the US represent data taken across very different time intervals with some sites starting years in the 1980s and others starting in 2023.  With the large changes in PM and its constituents across these decades the years that go into the temporal average will provide a very different picture of US air quality.

We add Section 3.1 to show that interannual variability is much smaller than other sources of uncertainty in this dataset.

In addition, there is no information on the sampling frequency between these years.  Although a total number of observations is given this is not sufficient information for important questions such as whether the instrument at a given site had measurements throughout the year or only certain parts of the year, or whether the measurements were daily, 1 in 3 days, 1 in 6 days, etc.  Also there is no start and end year provided for species

other than PM. Looking globally we also see substantial differences in the temporal sampling. Almost all of the data in China begins after 2010 and a large portion of the data in India begin in 2023.

We add in more details of when the sampling occurred, and add in the variable indicating when the data started for each constituent as recommended.

As a result, any spatial maps of these '1986-2023 average' aerosol measurements is essentially uninterpretable. The observed averages cannot be compared to model simulation data in any meaningful way although the authors include such a comparison against a global climate model.

We add Section 3.2 which shows that interannual variability is a small uncertainty in this dataset.

In addition to the .csv file with average measurements at each station, the authors have provided the temporally averaged observed data as spatially averaged gridded data over the model grid (~ 2° x 2°). I think the spatially averaging decreases the value of the dataset and I think these spatial fields do not need to be included with the data package for ESSD. For example, in the PM25 gridded obs file there is a grid cell with 261 different stations and 725,414 observations and another grid cell with 1 station and 3 observations. Yet these two 'observed' values are treated equally in the evaluation of the model output although they are vastly different in their representation of a 'true' air quality for their respective locations.

We provide this additional dataset in case users want it, but the data is also provided at the original resolution if the user prefers that.

In summary, why not provide the data at the original temporal and spatial resolution? This would allow the individual researcher to decide how to best spatially and temporally aggregate the original data for their application.

We do this, as indicated in the data availability section of the ACPD paper. We cannot post the data however, but it is available by request.

*Review of the data files*

*We fix all of the editorial comments, which make the paper and dataset much more useful.*

- The modelfiles.zip file contains a modelfiles.tar as well as the untarred files. Why the duplication?
- AEROMPdata.csv
  - All of the column headings need to be defined with units. The last columns in the file are missing column headings altogether.
  - The AROMAPdata.csv only has PM min and max year. Temporal information should also be included for the other species. When the PM measurement is missing there is no min and max year so there is literally no temporal record for those obs. Even when there is a PM measurement available I assume the temporal coverage can be quite different from species to species, so only including a min and max year is insufficient.

- - - - - The .csv includes negative longitude values as well as values up to 346.  The measurement location information should be harmonized so that values either range from 0-360 or fall with +/- 180.
      - There are records in the AEROMAPdata.csv where the PM min year is 0 and one record where the PM max year is 2167.
      - There are locations in Brazil and the islands off the coast of Morocco in the PM2.5 gridded obs file that are not in the AROMAPdata.csv file.
  - The .nc files
    - There needs to be more meta data than what is provided in the netcdf 'header' information or the README text file.  All variables should be clearly defined: gw, dep, source, conc, depmonth, concmonth, finconcmonth, coarseconcmonth.
    - The README file should also include the temporal averaging, i.e., 1986-2023.
    - The README includes this statement "Fine is PM2.5 and coarse is Pm10-Pm2.5, and  if not designated it is total PM10.".  The reference to 'not designated' is too vague.  I assume you are saying the variable 'conc' is PM10?  Are there other variable names in the other files other than 'conc'  that also refer to PM10?

**Technical Corrections**

*Main paper*

-
  -
    - Title should include the temporal coverage (1986-2023)
    - Title: Remove 'the'
    - Line 15: Should be "Raleigh"
    - Line 76: To keep this sentence parallel and avoid using 'impact' twice in  your list, recommend changing this to "...impact surface air quality, change surface albedo of snow and ice when deposited, and modulate..."
    - Line 82:  I disagree that the paper provides "a methodology".  The example evaluation has a lot of choices that are very specific to your model application.
    - Line 90: Recommend rewording with a more active voice. Also the qualified 'Recent' is probably not needed before "IPCC reports"
    - Line 103: Please add what "aerosol effects" is referring to in this sentence.
    - Line 157: formatting issue
    - Line 160: subscript needed
    - Line 182:  Please state explicitly where the time period information is included.  I had to hunt for it before finding it in the AEROdata.csv

file.  The temporal information is also inadequate in the .csv file as discussed above.

- Line 201: formatting issue
- Line 203:  Include the range of dates of the observations.
- Line 220:  Using 2010 emissions will certainly impact your model's ability to accurately estimate air quality in 2013-2015.  There are substantial changes in global air quality from 2010 -2015, for example decreasing emissions from industrial sources in China beginning in 2013.
- Line 227: formatting issue
- Line 228:  What is meant by "present day" in this context?  The previous paragraph refers to 2010 emissions for 2013-2015 simulations.
- Line 240:  What is meant by 'its importance can be isolated'
- Lined 241: O< should be OC
- Line 247:  Sentence beginning with 'Assumptions...' is awkwardly worded, particularly the parenthetical statement and could use some word smithing.
- Line 262:  Awkwardly worded.  How do particles 'interact with photochemistry'
- Line 282: Sentence beginning with 'We do not choose..' should be edited for clarity, for example breaking it into 2 or more sentences.
- Line 289: one-to-one
- Line 303:  What differences is this referring to, i.e., different between what two things?
- Line 312:  Why is the lat/long data location so imprecise in some cases?  In what geographic region does this occur?
- Line 314: Please reword sentence beginning 'Because of..' for clarity.
- Line 323:  What is meant by 'most' of the observations and 'most' of the stations?  Compared to what baseline?
- Line 324:  Please restate the temporal coverage (1986-2023)
- Line 324: You are not provided annual averages which implies you have separate averages for each year.  Rather you are providing an average over 1986-2023 of all available data.  At some sites you may not even have data for a single full year.
- Line 327:  Missing parentheses.
- Line 330: Because these are temporal averages I disagree "this dataset presents a huge increase in the amount of data available to the aerosol modelling community"
- Line 333:  These are not annual averages.  The observations span decades and there is no information on whether some of the sampling is only done during certain times of year.
- Line 337:  'there could be differences in the model-data comparison because of the time period discrepancies.'   Since you are not

matching the model values in time to the observations, it is not that there 'could' be differences, there most certainly ARE differences. It is hard for me to understand what can be learned from the evaluation study when the emissions and observations are so mismatched.

- Line 346: Suggest rewording to avoid the repeat of 'highest modelled values' and 'high concentration values'
- Line 348: "These discrepancies over India and China could be due to errors in the input emissions datasets or the aerosol transport modelling, or to differences in the time periods covered: the observations are more recent while the assumptions for the emissions are for the year 2010." Yes this is a huge discrepancy given the trends in PM in these areas between 2010 and 2023. It is hard to see what other conclusions can be drawn from this comparison because of this mismatch
- Line 354: This is a valuable comparison across observation sources. It would be valuable to spend more of the paper summarizing and validating the different sources of observed data.
- Line 369: An r value of .25 (r^2 = .06) does not support the statement that the model is 'roughly able' to capture spatial patterns.
- Line 383: Please reword sentence beginning "As a proxy for dust" for clarity
- Line 400: "NO3- aerosol particles compared against available observations show that over 2 orders of magnitude, the model results are able to simulate the spatial variability (Fig. 4k and l). Note that here, we have multiplied the simulations by a factor 0.5 in order to achieve a good mean comparison, as indicated by Vira et al. (2022). " I strongly disagree with the approach of scaling a model result to 'achieve a good mean comparison' and then claiming that the model is able to simulate observations. The Vira et al. paper is documenting a model bias, not suggesting that model values should be scaled by a single factor across space and time.
- Line 408: The use of PM6.9 notation is non-standard and could create unnecessary confusion. Other air quality models calculate a cutoff for every data point based on the composition of the particles at that time and place.
    - For an example from the Community Mulitscale Air Quality (CMAQ) here is the equation to go from Aerodynamic diameter cutoff (i.e. PM10) to Stokes diameter cutoff: https://github.com/USEPA/CMAQ_Dev/blob/24c0840315978f94541d8b6288163e7a54c8694d/CCTM/src/aero/aero6/aero_subs.F#L2737
    - D_stokes = D_aerodynamic * (Rho0 / RhoP)^(1/2) * (Cc(D_aerodynamic)/Cc(D_stokes))^(1/2)
    - Rho0 is assumed to be 1.0 g cm-3

- - RhoP is the particle density (i.e., there are different RhoP values for different sources such as dust and sea spray)
  - CMAQ output is reported as PM10 (aerodynamic diameter space) rather than in Stokes diameter space and PM10 is the standard nomenclature for air quality modeling and evaluation against observations.

In the resubmission to ACP we considered implementing this, but we were on the fence and so left it as it was. For this revision we include this edit.

- - Line 417: Please reword sentence beginning "Again we are including new.." for clarity.
  - Line 431: Please reword sentence beginning "Similarly,"
  - Line 437: "For the NO3- in aerosol particles, similar to the PM2.5 size, the particles were multiplied by 0.5 to better match the observations following Vira et al (2022)." Same comment as above. It is not appropriate for a model evaluation study to scale model results and them proceed to describe model 'performance' of the scaled values. Rather than this approach you could summarize the evaluation in Vira et al (2022) and compare it to the biases you see in your study.
  - We include this in the new version.
  - Line 488: "The fidelity of the annual means provided by this study will depend upon the ability of the measurement networks to capture the observed multi-decadal increases and decreases of emission that vary between source regions and sectors (Quaas et al., 2022)."
    - Again you are not providing annual means but a multi decadal mean. There is no way for the single average to capture trends in the data.

*Figures*
- all figures (including the supplement) should include the time period of the averaging in the caption, i.e., 1986-2023.
- Figure 1: The sources of the data should not be included in the figure caption but in the main text or as supplemental information.
- We chose to include it both in the supplemental information and in the figure caption: otherwise it would fill a whole page.
- Figure 2a: The colors of the gridded observations do not show up due to the outline of the colored dots. It would be helpful if you could add a column that is just the obs data (and remove black outline of the dots) then have middle column be just the model values.
- The reviewer missed that we had figures in the supplement of just the modeled values.
- Figure 2a: You include the comment "see Supplemental dataset for more details". Since you have multiple supplemental datasets, please be explicit about where this information is located, i.e., name of file, where to find it in the file.

- o There is only one supplemental dataset for the submitted manuscript.
- o Figure 2d: It is difficult to glean much from a scatter plot with over 7,000 points. It would be more helpful to see separate density scatter plots for each of the 7 regions where the color of the plot indicates the density of the points. This would allow the reader to more easily see consistent biases and correlation in each region.
- o We add in regional averages.
- o Figure 4: Same comment as Figure 2 for the observation overlays on the model maps.
- o All figures: Using the log scale for model vs obs emphasizes smaller values and condenses the largest values. However from a health and ecosystem standpoint the largest values are of greatest interest and there is not much need to know the difference in model performance for .1 ug/m3 vs .01 ug/m3. Strongly suggest not plotting on log scale. Also recommend a density scatter plot when # points exceeds several hundred.
- o However, for the aerosol-cloud interaction and impact on climate, 0.1 versus 0.01 really matters (Carslaw et al.), and the relative amount of different sources matter, so we think it is important to show on a log log graph.
- o Figure 7. Include in the caption information the time period you are considering for the % coverage.
- o This is over ANY time period (since interannual variability is a small part of the signal in this dataset).

**Citation**: https://doi.org/10.5194/essd-2024-1-RC1

Reviewer 2:

**General comments**

This manuscript includes a comprehensive collection of aerosol composition data and provide a global overview to be compared with an earth system model. This is certainly an important effort since there is little work in the global perspective to include a complete analysis of the aerosol composition, there is especially lack of global information on the spatial distribution of coarse aerosols. A synthesis of data from a lot of different sources is great effort by the authors,

However, I find the lack of transparency of the data sources and the choice of data for the model comparison concerning. It seems a bit arbitrary which data has been selected and a reasoning for why you have selected those you have. There is a mixture of data from shorter campaign periods, others are compilations of data from regional programmes with long time series, and these are not necessarily comparable when everything is aggregated. There are no references or acknowledgement of the data from the large regional and global data repositories in the manuscript, which I presume quite a lot of the data are collected from, I.e. CAPMoN, CASTNET, EANET, EMEP, IMPROVE, INDAAF and GAW. I find it quite troublesome that the primary data repositories are not acknowledged or referred to better.

All the data that could be found was included. We clarify this. All the citations were in the supplemental dataset, so it's possible the reviewer didn't find this. We added doi information as well, and this information is in the supplemental bibliography.

Further it is not clear why (and maybe how) the observations have been gridded/aggregated. There can be huge differences in concentrations between sites and between years, thus how well are the average concentrations representative for the grid? I.e. if you have 10 urban sites and one regional within the same grid, and some data for 1 year and others for 20 years? The authors should try to choose representative sites for the grids.

For this paper we do a simple approach just to make plotting easier, but the underlying data is available if someone wants to aggregate/grid it differently.

I also find the selected time periods for the comparison between model and observations troublesome. The model results are for the period 2013-2015 with emissions from 2010, and this is compared to data from a very large time period, 1986-2023. I don't understand the usefulness of comparing this. The temporal changes in atmospheric chemistry have been large during this time period, with different trajectories for the various regions. Further some years probably have much more data than others. I think you should only use observational data for the same period ±2 years as the model (e.g 2010-2015), and not use data from all the way back to 1986.

The reviewer asserts that interannual variability is an important source of error, and we add Section 3.2 to show that this is not the case for this dataset, although it may be the case for another dataset (the reviewer supplies no citations to support this.)

**Specific comments**

As mentioned, the references to the data used need to be improved. An example of poor referencing is Alastuey 2016. This is a European campaign of dust measurements (2012-2013), but in the AEROMAP datafile this reference is used for data from Montseny only and for a longer period (2012-2017). Data from EMEP is referred to as Tørseth et al., 2012, that is ok, though newer reviews have been done. But more importantly, the data is collected from the data repository EBAS, and not from the paper itself. The reference to EBAS is lacking in the paper.

We add the reference to EBAS, and we include the dois for the data from this source. We use the citations supplied to us by the PI on the dataset (Alastuey et al., 2016, as an example).

It is stated in the csv file, but this is very much hidden. Further, quite a lot of data from EMEP is not included in the study of some reasons.

We include all data that we can download from the EBAS dataset for EMEP.

The Asian EANET data is available from the EANET data repository (https://www.eanet.asia/) and should not be references to Tørseth et al., 2012.

Unfortunately the EANET dataset was not available from the eanet website, but was available from the EBAS dataset. We fixed the citation.

Another example is from Canada where the link to the data is not working: https://data.ec.gc.ca/data/air/monitor/national-air-pollution-surveillance-naps-program/Data-Donnees. These data from Environmental Canada I assume?  If so, https://open.canada.ca/en/open-data is probably more appropriate, or maybe not -it seems like you lack a lot of chemistry data from Canada? I only did some test of references and links. There is a need for more checks by the authors.

We include all the data that we could find from this website.  The website works for us, so we do not know the problem. If there is more data, please provide more detailed citations and websites, as we would be happy to include.

- line 135-136: I agreed certainly with the open data policy i.e. that monitoring networks and those using the data follow the FAIR data policy. However, I don't see that this paper contributes so much to that. FAIRness of data includes proper links (i.e. with DOIs) to the data providers, and not compilations on files where there is poor tracking of the origin.
- We include coauthorship for PIs that want it, and citation of all datasets as appropriate.

- Line 167. Following upon the comment above. AEROMAP database should not be defined as a the global database without looking into the data policy for the original data repositories. Duplicating repositories is not a good way forward. Certainly, it is great that you made the data you have used available, but the way it is written you suggest the reader to go to AEROAMAP to collect aerosols data in the future. This is not sustainable.
- We agree AEROMAP is not sustainable.  We hope the GHOST dataset will be sustainable, but it is not as complete as our dataset, so want to make it available for others in the future.

- 146-147. Do you constrain the model by observations? It is rather a comparison and not an adjustment of the model output. I don't find any text on how the model has been constrained by the measurements. In the next sentence the constrain is rather referred to as getting a full aerosol budget using tracers to estimate sea salt (Na) and dust, (Al) but constraining the PM budget that is something different? It would have been nice to show a spatial distribution plot with relative contribution of i.e. SIA, organics, sea salt, and dust to PM10 and PM2.5 if possible.
- We add this to the supplemental figures.
- 179-181, 324-325 and Fig 1 Not clear what is defined as number of observations. For e.g. a time series of PM10 is 10 years. Is it then 1,10 or 520 if weekly samples, but had it been hourly data it would have been 87 600 data points. Though how can you have 10^6 observations of OC? In Europe (EMEP program) there are ca 15 sites measuring OC and 10 measuring Al in aerosols depending on which year and most of these measurements are weekly and it does not sum up to millions of observations.  It would make more sense to illustrate the number of annual datasets you use, or the number of comparison points (grids) with model. Which I assume is what is given in table S4.

- This can be done many ways: We include the number of data points, and the number of station datasets, and try to clarify this.

  I find it very strange that you have almost the same number of sites for EC, OC and Al compared to SO4 which certainly is measured at many more sites globally. Though you are maybe picking those sites which have all the constituents and not only one of them? The number of sites looks reasonable in Fig 1b and c, but I don't understand the number in Fig1e. How can it by number of sites "for each 2x2 grid box is shown as a dotted line"? It must be a different number for each gird box depending of the spatial distribution of the sites.
- We clarify in the figure caption.
- 307-308 and repeated at 342. I don't understand the reasoning that the high number of sites will make the figure unreadable. The reason for gridding is to rather better compare with the model output?
- We simply can't see individual datapoints if we plot each station separately.

**Technical corrections**

Thank you for these corrections, and we include them.

- Figure caption fig 1: the link https://app.cpcbccr.com/ccr/#/caaqm-dashboard-all/caaqmlanding/data does not work
- Line 227: Missing space between BC and follow
- 248: P is phosphorus?. The sentence in the parentheses is a bit strange
- Line 241 written O< instead of OC or OM

**Citation**: https://doi.org/10.5194/essd-2024-1-RC2

---

## Author Response (AR2)

Thanks to the editor for carefully reviewing our responses and the manuscript. We make the suggested edits as requested by the editor as they help clarify the text. Comments from the editor are in red, our responses are in black.

Thank you for the substantial revision of the manuscript and for addressing all reviewer comments. After reviewing your responses and the revised manuscript, I am pleased to accept it for publication, pending the following minor additions and corrections.

Line 92-95: It should be mentioned that the estimate is at surface.

Good point. The sentences are rewritten to:

"Climate models without ammonium nitrate aerosols omit ~10% of the globally averaged surface concentration of aerosol particles in both $PM_{2.5}$ and $PM_{10}$ size fractions, with up to 50% of the surface concentrations not included in some regions."

Line 148 & 1489: Update to the published paper, so Thornhill et al. 2020 -> Thornhill et al. 2021a.

Thank you, good point: we fix this throughout the paper.

Line 209-210. A reference for the choice of OM to OC ratio of 1.8 would be encouraged since the ratio is quite variable. (Ratio given as well at line 318-319).

Good point. We add the the following citations to justify our choice in 209-210 Aiken et al., 2008; Font et al., 2024; Turpin & Lim, 2001.

Lines 318:321: "For example, OM is assumed to be 1.8 times OC if OC measurements are available but not OM measurements. Different ratios of OM to OC appear in the literature, but 1.8 appears to be the best average for a mixture of aged and fresh plumes (Aiken et al., 2008; Font et al., 2024; Turpin & Lim, 2001).

Line 496-498: Overestimation of SO2 could also be due to too slow reaction to sulphate or too slow deposition. Consider expanding on this point.

Good point, there are a lot of reasons for model errors, so we add in these examples, and try to make it clear there are many reasons for model errors:

501-503 "In addition, these discrepancies could be due to an error in the aerosol transport or chemical modelling, such as incorrect reaction rates or deposition rates or alternative

due to the differences in the time period: the observations are more recent while the assumptions for the emissions are for the year 2010 (Quass et al., 2021)."

Line 855-858: Update to the final published paper of Bowdalo et al. (2024)

Thank you. We do this.

Line 945: Include journal for Fasullo et al.

Thank you for catching this.

Line 1124: Include journal for Kok et al.

Thank you for catching this.

Line 1165: Update to the final published paper of Li et al

Thank you, we do this.

Line 1335: Correct page numbers for Neff et al.

Thank you, we correct this.